# Atrial-like cardiomyocytes from human pluripotent stem cells are a robust preclinical model for assessing atrial-selective pharmacology

Harsha D Devalla[1],[*], Verena Schwach[1], John W Ford[2], James T Milnes[2], Said El-Haou[2], Claire Jackson[2], Konstantinos Gkatzis[1], David A Elliott[3], Susana M Chuva de Sousa Lopes[1],[4], Christine L Mummery[1], Arie O Verkerk[5] & Robert Passier[1],[**]

## Abstract

Drugs targeting atrial-specific ion channels, $K_v1.5$ or $K_{ir}3.1/3.4$, are being developed as new therapeutic strategies for atrial fibrillation. However, current preclinical studies carried out in noncardiac cell lines or animal models may not accurately represent the physiology of a human cardiomyocyte (CM). In the current study, we tested whether human embryonic stem cell (hESC)-derived atrial CMs could predict atrial selectivity of pharmacological compounds. By modulating retinoic acid signaling during hESC differentiation, we generated atrial-like (hESC-atrial) and ventricular-like (hESC-ventricular) CMs. We found the expression of atrial-specific ion channel genes, *KCNA5* (encoding Kv1.5) and *KCNJ3* (encoding $K_{ir}$ 3.1), in hESC-atrial CMs and further demonstrated that these ion channel genes are regulated by COUP-TF transcription factors. Moreover, in response to multiple ion channel blocker, vernakalant, and $K_v1.5$ blocker, XEN-D0101, hESC-atrial but not hESC-ventricular CMs showed action potential (AP) prolongation due to a reduction in early repolarization. In hESC-atrial CMs, XEN-R0703, a novel $K_{ir}3.1/3.4$ blocker restored the AP shortening caused by CCh. Neither CCh nor XEN-R0703 had an effect on hESC-ventricular CMs. In summary, we demonstrate that hESC-atrial CMs are a robust model for pre-clinical testing to assess atrial selectivity of novel antiarrhythmic drugs.

**Keywords** arrhythmias; atrial cardiomyocytes; atrial fibrillation; COUP-TF; ion channels

**Subject Categories** Cardiovascular System; Pharmacology & Drug Discovery; Stem Cells

## Introduction

Atrial fibrillation (AF) affects over 33 million people globally (Chugh *et al*, 2014) and is characterized by irregular atrial rhythm leading to a decline in atrial mechanical function. Untreated AF increases the risk of life-threatening complications such as stroke or heart failure (Wang *et al*, 2003; Marini *et al*, 2005). Current treatment options for rhythm control in AF include interventional therapy such as ablation or pharmacotherapy with antiarrhythmic drugs. The latter is the preferred treatment of early AF in individuals who prefer non-invasive treatment and as a follow-up therapy postelectrical cardioversion, to prevent recurrence of AF (Wann *et al*, 2011). However, existing antiarrhythmic agents lack atrial selectivity and pose the risk of inducing undesirable cardiac events, such as ventricular proarrhythmia (Dobrev & Nattel, 2010).

In order to overcome this limitation, pharmaceutical industry has initiated design and development of compounds aimed at atrial-specific targets (Li *et al*, 2009; Milnes *et al*, 2012). The notorious difficulty in obtaining human cardiomyocytes (CMs) and propagating them in culture has precluded their use from many drug screening assays and instigated the use of alternative preclinical models. However, many of the current preclinical screening assays used in the identification of atrial-selective compounds are performed using either non-cardiac recombinant cell lines expressing a non-native ion channel or animal models. Both these models may not accurately represent the ion channel composition as well as physiology of a human CM and therefore have limitations in predicting drug responses on the human heart.

Human pluripotent stem cell-derived CMs (hPSC-CMs) offer a human-based, physiologically relevant model system for drug discovery and development strategies. Despite the suitability of these cells for cardiotoxicity testing and safety pharmacology (Braam *et al*, 2010; Navarrete *et al*, 2013), their application in validating

1 Department of Anatomy & Embryology, Leiden University Medical Center, Leiden, The Netherlands
2 Xention Ltd, Cambridge, UK
3 Murdoch Childrens Research Institute, Royal Children's Hospital, Melbourne, Vic., Australia
4 Department for Reproductive Medicine, Ghent University Hospital, Ghent, Belgium
5 Heart Failure Research Center, Academic Medical Center, University of Amsterdam, Amsterdam, The Netherlands
*Corresponding author. Tel: +31 71 5269528; Fax: +31 71 5268289; E-mail: h.d.devalla@lumc.nl
**Corresponding author. Tel: +31 71 5269359; Fax: +31 71 5268289; E-mail: r.passier@lumc.nl

 

novel drug candidates for AF requires cultures enriched in atrial-like CMs. Current protocols for cardiac differentiation of hPSCs result in heterogeneous pools of CMs consisting predominantly of ventricular-like cells with a small percentage of atrial-like and nodal-like cells (Blazeski *et al*, 2012). Based on substantial evidence from *in vivo* and *in vitro* studies (Niederreither *et al*, 2001; Hochgreb *et al*, 2003; Gassanov *et al*, 2008; Zhang *et al*, 2011) indicating a role for retinoic acid (RA) in atrial specification, we hypothesized that RA would drive mesodermal progenitors from PSCs toward an atrial fate.

In the current study, we show that transcriptional and electrophysiological properties of human embryonic stem cell-derived atrial CMs (hESC-atrial CMs), generated by modulating RA signaling, closely resemble that of native human atrial CMs. We also observed that transcription factors, COUP-TFI (NR2F1) and COUP-TFII (NR2F2), are robustly upregulated in response to RA during directed atrial differentiation. Short hairpin RNA (shRNA)-mediated knockdown and chromatin immunoprecipitation (ChIP) of COUP-TFs identified that they regulate atrial-specific ion channel genes *KCNA5* (encoding $K_v1.5$) and *KCNJ3* (encoding $K_{ir}3.1$). Furthermore, hESC-atrial CMs express atrial-selective ion currents, $I_{Kur}$ as well as $I_{K,ACh}$, and also respond to pharmacological compounds targeting ion channels that conduct these currents ($K_v1.5$ and $K_{ir}3.1/3.4$, respectively).

Collectively, our data identify a key role for COUP-TF transcription factors in RA-driven atrial differentiation and also demonstrate that hESC-atrial CMs are a robust model for predicting atrial selectivity of novel pharmacological compounds during preclinical development.

# Results

### Treatment of differentiating hESCs with RA promotes atrial specification

A cocktail of cytokines (Fig 1A) was used to initiate cardiac differentiation in *NKX2-5-eGFP/w* hESCs as described previously (Ng *et al*, 2008; Elliott *et al*, 2011). To direct differentiating hESCs toward an atrial phenotype, timing and concentration of treatment with RA were carefully optimized (Supplementary Fig S1A–C). We hypothesized that specification of CM subtypes *in vitro* occurs post-mesoderm formation and prior to the onset of cardiac progenitor stage. Accordingly, embryoid bodies (EBs) were supplemented with RA from day 4, just after the transient expression of early cardiac mesoderm marker, *MESP1,* until day 7, a time point at which key transcription factors such as NKX2.5, GATA4 and MEF2C important for commitment and specification of cardiovascular lineages are activated (Supplementary Fig S1A).

Adding low concentrations of RA (1–10 nmol/l) from day 4 to 7 enhanced cardiac differentiation as assessed by the percentage of GFP$^+$ cells at day 15 (Supplementary Fig S1B). On the other hand, treatment with high concentrations of RA (1 μmol/l) in the same time window resulted in GFP$^+$ EBs with reduced expression of the ventricular specific myosin gene, *MLC2V* (Supplementary Fig S1C). Therefore, treatment of differentiating hESCs with 1 μmol/l of RA from day 4 to 7 was considered to be most suitable for driving atrial differentiation. As a control, every experiment included parallel

differentiating cultures treated with 0.002% DMSO (the final concentration in RA-treated cultures) from day 4 to 7 (Fig 1A).

Morphologically, RA-treated EBs were similar compared with control EBs (Supplementary Fig S1D), and contractile GFP$^+$ areas were observed in both groups at day 10 (Fig 1B). Flow cytometry analysis of GFP expression at day 15 revealed a decrease in the proportion of NKX2.5-expressing cells upon treatment with RA. Sixty-five percent of all cells expressed GFP in control differentiation, while only 50% cells in RA-treated differentiation were GFP$^+$ (Fig 1C and Supplementary Fig S1E). These data are consistent with earlier reports in zebrafish embryo which demonstrated that exposure of anterior lateral plate mesoderm to RA signaling restricts the size of the cardiac progenitor pool (Keegan *et al*, 2005). Also, EBs treated with RA during differentiation displayed faster beating frequencies upon differentiation (Supplementary Fig S1F). Finally, immunofluorescence analysis of EBs from both control and RA-treated differentiations confirmed that the contractile GFP$^+$ areas expressed both NKX2-5 and the myofilament marker, ACTN2 (Supplementary Fig S1G).

### Transcriptional profiling of CMs from RA-treated differentiations reveals an upregulation of atrial and downregulation of ventricular markers

To study gene expression in CMs resulting from control and RA-treated conditions, cells were sorted on the basis of NKX2-5-eGFP. GFP$^+$ and GFP$^-$ fractions from control (CT+/CT−) and RA-treated EBs (RA+/RA−) isolated at day 31 post-differentiation were assessed by microarray and quantitative PCR (qPCR). Expression of contractility genes, *TNNC1* and *TNNT2,* calcium handling genes, *RYR2* and *ATP2A2*, and cardiac transcription factors, *MEF2C* and *NKX2.5* were enriched in both CT+ and RA+ pools compared with CT− and RA− populations, indicating efficient purification of CMs (Fig 1D). A heat map of GFP$^+$ and GFP$^-$ samples also demonstrates strong correlation with each group (Supplementary Fig S2A). Microarray analysis demonstrated upregulation of atrial markers such as *SLN*, *HEYL*, *PITX2* and *NPPA* (Fig 1E), while ventricular markers such as *MYL2*, *IRX4*, *HAND1* and *HEY2* were downregulated in RA+ CMs (Fig 1F). Measuring expression levels of selected targets by quantitative qPCR further validated the microarray data in which expression of *NKX2-5* and *TNNT2* was significantly higher in GFP$^+$ fractions (Fig 2A). qPCR also confirmed upregulation of atrial and downregulation of ventricular transcripts in RA+ CMs (Fig 2B and C).

In order to compare the expression profile of CT+ and RA+ CMs at day 31 with that of human heart, we included atrial and ventricular tissue samples of a 15-week-old fetal heart for microarray analysis. A total of 151 genes showed increased expression of more than twofold in CT+ group compared to RA+. Thirty-two percent of these genes (49 out of 151) were preferentially expressed in human ventricles, whereas only 8% (13 out of 151) could be identified in the group of genes that were enriched in the human atria (Fig 2D). On the other hand, 292 genes showed increased expression of more than twofold in RA+ group compared to CT+. Thirty-one percent of the genes (92 out of 292) with enriched expression in RA+ group were preferentially expressed in the human atria, while a mere 8% (23 out of 292) of these genes were expressed in the human ventricles (Fig 2E). Gene lists in Venn diagrams (Fig 2D and E) are

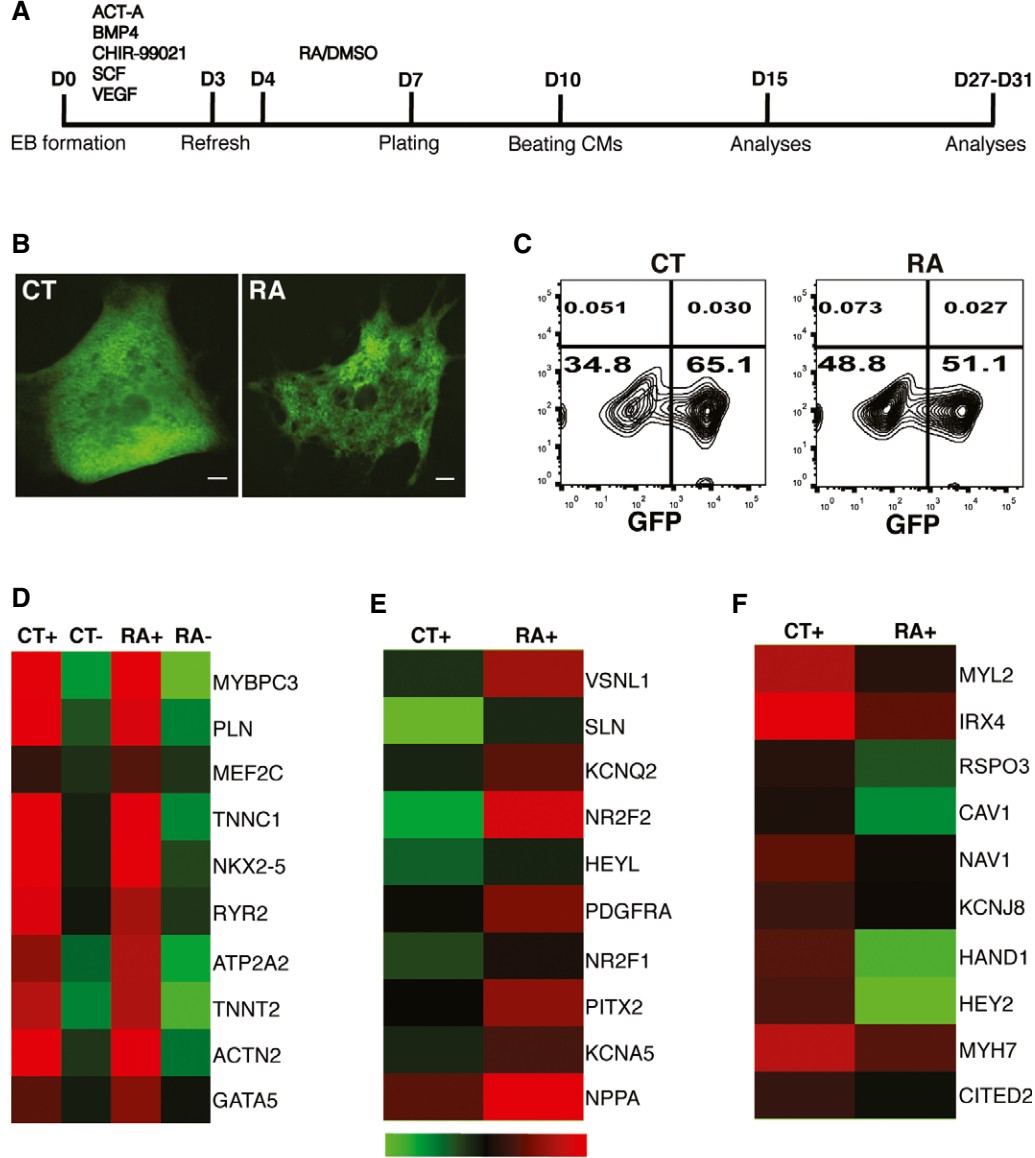

**Figure 1. Treatment of differentiating hESCs with RA promotes atrial specification.**

A   Schematic of the cardiac differentiation protocol. Beating embryoid bodies (EBs) were observed at day 10. Differentiation efficiency in each experiment was assessed by flow cytometry (FC) for GFP at day 15. Further characterization of EBs derived from control (CT) and RA-treated (RA) cultures was carried out by transcriptional or functional analysis between days 27 and 31.

B   GFP+ EBs derived from CT and RA cultures at day 10; scale bar: 100 μm.

C   Representative FC plots depicting percentage of GFP+ cells obtained at day 15, from CT and RA cultures in a typical experiment.

D   Heat map demonstrating enrichment of cardiac genes in GFP+ fractions (CT+, RA+) compared to GFP− fractions (CT−, RA−) at day 31.

E, F   Heat map of a select list of genes (E) upregulated and (F) downregulated in RA+ compared to CT+ at day 31. Fold change > 2.

included in Supplementary Table S1. Pie charts illustrating the cellular localization and molecular function of genes enriched in CT+ and RA+ groups are shown in Supplementary Fig S2B and C. Gene ontology (GO) analysis of microarray data was performed with ConsensusPathDB-human, and terms satisfying a cutoff of $P < 0.01$ were considered enriched. GO terms related to four major classes, cardiovascular development, muscle development, developmental process and adhesion, were overrepresented in both groups (Fig 2F). In the category of cardiovascular development,

GO terms such as appendage development, cardiac atrium development and cardiac septum development were enriched in RA+ CMs while cardiac ventricle development, ventricular septum formation and heart trabecular formation were enriched in CT+ CMs (Supplementary Table S2). A detailed list of GO terms in each category and the genes are included in Supplementary Table S3.

Therefore, the transcriptional profile of RA+ CMs suggested a fetal atrial-like gene expression pattern compared with control CMs, which expressed higher levels of ventricular transcripts.

**Figure 2. Transcriptional analysis of CT+ and RA+ CMs.**

A–C   qPCR of selected transcripts at day 31 to validate (A) enrichment of cardiac markers in GFP+ fractions against GFP− fractions, (B) upregulation of atrial and (C) downregulation of ventricular genes in RA+ compared to CT+ (n = 3).

D, E   Venn diagram to illustrate overlap of gene lists upregulated (UP) in CT+ CMs (D) and RA+ CMs (E) with genes expressed in atria and ventricles of 15-week-old fetal heart. Red square indicates higher overlap of CT+ with gene list of fetal ventricles or higher overlap of RA+ with gene list of fetal atria.

F   Pie chart illustrates major classes of gene ontology terms enriched in gene lists upregulated in CT+ and RA+ CMs.

Data information: Data are presented as mean ± SEM. In (A–C), *P < 0.05, **P < 0.01, ***P < 0.001 by unpaired t-test. In (A), for TNNT2, P = 0.0001 for CT− against CT+ and P = 0.0002 for RA− against RA+; for NKX2.5, P = 0.00005 for CT− against CT+ and P = 0.00007 for RA− against RA+. In (B), P = 0.0006 for NPPA and P = 0.0002 for PITX2. In (C), P = 0.02 for HEY2 and P = 0.007 for IRX4.

## AP characterization of CMs generated from RA-treated differentiations establishes their atrial-like phenotype

To study the electrical phenotype of CMs treated with RA during differentiation, we measured APs of dissociated single cells using the patch-clamp method (Fig 3). Representative APs stimulated at 1 Hz are shown in Fig 3B. The AP of a CM treated with RA during differentiation (RA) depicts an AP with a fast phase-1 repolarization and a plateau phase with a more negative potential compared to a control (CT) CM (Fig 3B). Average RMP (Fig 3C)

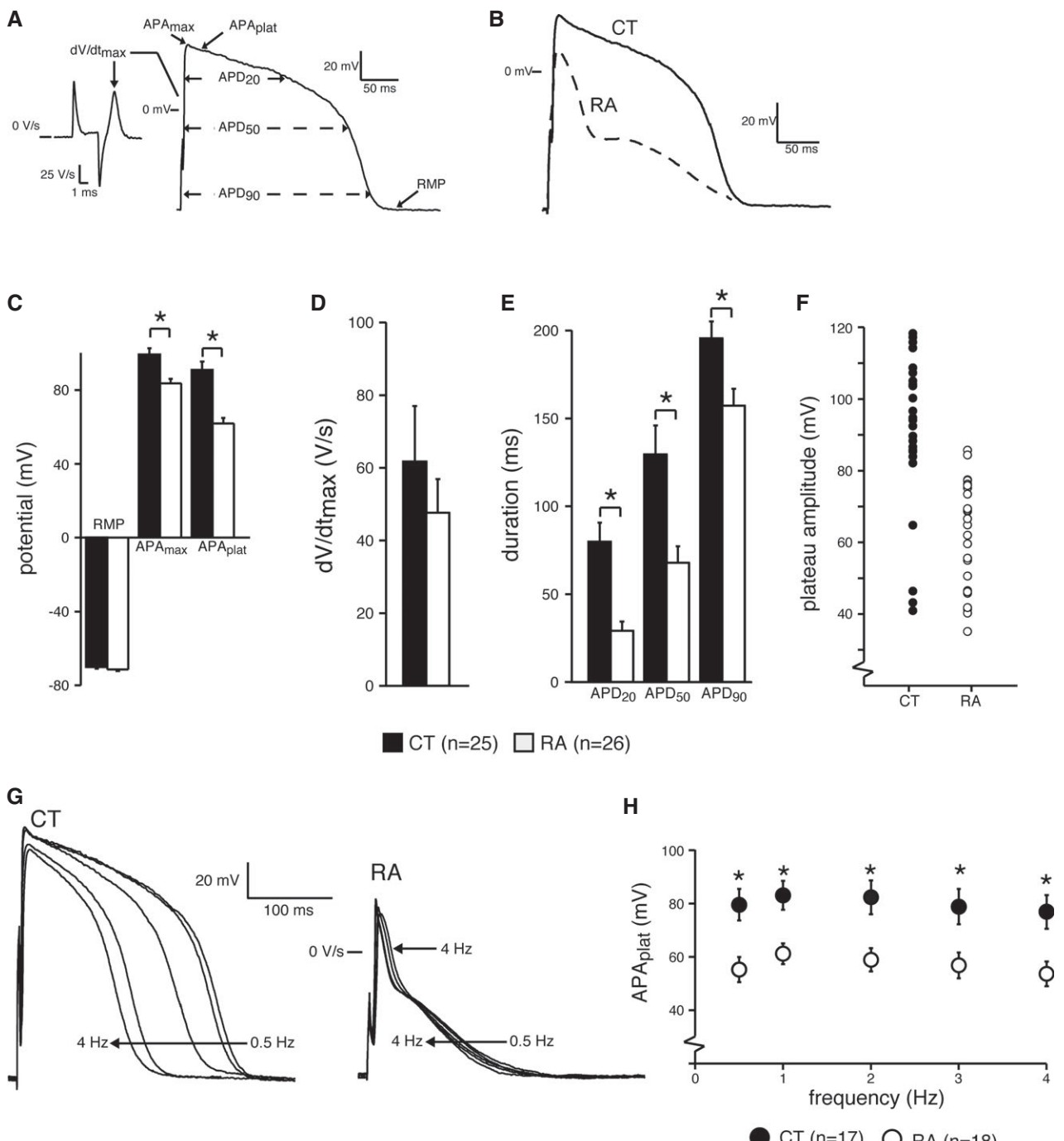

**Figure 3. AP characterization of CMs generated from control and RA-treated differentiations.**

A    AP illustrating the analyzed parameters.
B    Representative APs of day 31 CMs from control (CT) and RA-treated (RA) groups at 1 Hz.
C–E  RMP, $APA_{max}$ and $APA_{plat}$ (C), $dV/dt_{max}$ (D) and $APD_{20}$, $APD_{50}$ and $APD_{90}$ of CT and RA CMs (E).
F    Plot showing all measured $APA_{plat}$ values of CT and RA CMs.
G    Representative APs of CT and RA CMs at 0.5–4 Hz.
H    Average $APA_{plat}$ at 0.5–4 Hz. Please note that the AP differences in morphology are present at all measured frequencies.

Data information: Data are presented as mean ± SEM. *$P < 0.05$ by unpaired *t*-test or Mann–Whitney rank-sum test for (C–E). In (C), $P = 0.238$ for RMP; $P < 0.001$ for $APA_{max}$ and $APA_{plat}$. In (D), $P = 0.598$ for $dV/dt_{max}$. In (E), $P \leq 0.001$ for $APD_{20}$; $P = 0.009$ for $APD_{50}$; $P = 0.04$ for $APD_{90}$. Two-way repeated measures ANOVA followed by pairwise comparison using the Student–Newman–Keuls test for (H). *$P = 0.002$, 0.006, 0.003, 0.004 and 0.003, respectively, for comparison of $APA_{plat}$ between CT and RA groups at frequencies of 0.5, 1.0, 2.0, 3.0 and 4.0 Hz. AP = action potential; $APA_{max}$ = maximum AP amplitude; $APA_{plat}$ = AP plateau amplitude; $APD_{20}$, $APD_{50}$ and $APD_{90}$ = AP duration at 20, 50, and 90% repolarization, respectively; CMs = cardiomyocytes; $dV/dt_{max}$ = maximum upstroke velocity; RMP = resting membrane potential.

                                    

and $dV/dt_{max}$ (Fig 3D) did not differ significantly between the two groups.

In particular, APs of RA CMs had a significantly lower $APA_{max}$ (Fig 3C). They also repolarized faster resulting in significantly lower $APA_{plat}$ (Fig 3C) and shorter $APD_{20}$, $APD_{50}$ and $APD_{90}$ (Fig 3E). A scatter-plot of individual $APA_{plat}$ values clearly shows that the plateau amplitudes of individual cells in the RA group were typically < 80 mV, whereas those in the CT group were > 80 mV (Fig 3F). The AP differences between the two groups were also consistent when the cells were paced at higher frequencies (Fig 3G and H). Of 25 cells measured from the control group, about 80% (of 25 cells) displayed ventricular-like action potential properties while about 85% (of 26 cells) in the RA treatment group showed atrial-like action potential properties. We observed a very small percentage (< 1%) of nodal-like cells in both the groups.

The differences observed in AP duration and $APA_{plat}$ between RA and CT CMs closely matched the AP differences observed between atrial and ventricular CMs *in vivo* (Nerbonne & Kass, 2005). Taken together, gene expression signature and electrophysiological properties demonstrated that CMs treated with RA during differentiation displayed atrial-like phenotype (hereby referred to as hESC-atrial CMs), while control CMs resembled ventricular-like cells (hereby referred to as hESC-ventricular CMs).

## COUP-TFI and COUP-TFII are upregulated in response to retinoic acid, and their expression persists in differentiated hESC-atrial CMs

Transcriptional profiling experiments revealed that orphan nuclear receptor transcription factors, COUP-TFI (NR2F1) and COUP-TFII (NR2F2), are highly upregulated in hESC-atrial CMs. Based on previous reports by others indicating the involvement of COUP-TFs in RA signaling (Jonk *et al*, 1994; van der Wees *et al*, 1996), we postulated that these genes might play a central role downstream of RA during atrial differentiation. This hypothesis was further supported by atrial-specific expression of *Coup-tfII* in the mouse and severe atrial abnormalities observed in the loss-of-function mouse mutant (Pereira *et al*, 1999). A more recent study found that Coup-tfII regulates atrial identity in the mouse heart (Wu *et al*, 2013).

In order to determine the dynamics of COUP-TF expression following RA treatment, expression levels of both *COUP-TFI* and *II* were analyzed by qPCR at different time points and compared to control EBs. *COUP-TF*s were induced within 24 h of treatment with RA followed by dramatic increase in expression thereafter. A line graph plotting the relative mRNA levels of *COUP-TFI* and *II* between control and RA-treated groups shows striking differences, indicating that addition of RA induced the expression of these orphan nuclear receptor transcription factors (Fig 4A and B). The expression of *COUP-TF*s was maintained in differentiated CMs at day 31. *COUP-TFI* was expressed 20-fold higher, and *COUP-TFII* was enriched over 30-fold in hESC-atrial CMs compared to hESC-ventricular CMs (Fig 4A and B). Antibodies selectively binding to COUP-TFI or COUP-TFII were used to verify the expression of these proteins. While GFP$^+$ areas in hESC-ventricular CMs at day 25 showed relatively low expression of COUP-TFI and COUP-TFII, hESC-atrial CMs showed robust expression of these transcription factors (Fig 4C and D).

To confirm our *in vitro* findings, which identified high levels of COUP-TFs in hESC-atrial CMs, we sought to verify the expression of COUP-TFI and COUP-TFII in the human heart. Previous studies by others have reported preferential expression of *Coup-tfII* in the atrial myocardium of the mouse heart (Pereira *et al*, 1999), but no data are available for *Coup-tfI*. qPCR identified significantly higher mRNA levels of *COUP-TFI* and *COUP-TFII* in the atria as opposed to ventricles in both human fetal and adult heart (Supplementary Fig S3A and B). In accordance with the mRNA expression levels, COUP-TFI protein showed nuclear localization in the myocardium of the atrial chambers stained with TNNI3 (Supplementary Fig S3D and E) while no expression was detected in the myocardium of ventricles (Supplementary Fig S3F and G) or elsewhere in two of the analyzed hFHs at 12 weeks of gestation. Similarly, strong expression of COUP-TFII was observed in the TNNI3-positive myocardium of the atria (Supplementary Fig S3H and I), while no expression was found in the ventricular myocardium (Supplementary Fig S3J and K) of the hFH. Collectively, expression and histochemical analysis in human fetal hearts demonstrate that COUP-TFI and II are indeed expressed in the atrial myocardium of the human heart as observed in hESC-atrial CMs *in vitro*.

## COUP-TFs regulate expression of atrial-specific potassium channel genes, *KCNA5* and *KCNJ3*

To investigate whether COUP-TFs have an essential role in differentiated CMs, we used shRNAs to knockdown *COUP-TFI* or *COUP-TFII* in hESC-atrial CMs and studied whether they regulate atrial-specific ion channel genes. Lentiviral pLKO.1 constructs containing five different shRNA sequences each (Supplementary Table S4), for *COUP-TFI* and *COUP-TFII* (Supplementary Fig S4A–C), were tested in hESC-atrial CMs. Two *COUP-TFI*-shRNAs (#2; #4) and two *COUP-TFII*-shRNAs (#7; #10) gave efficient knockdown as assessed by qPCR (Supplementary Fig S4D) and were selected for further experiments. Transduction of *COUP-TFI*-shRNA or *COUP-TFII*-shRNA in hESC-atrial CMs resulted in 70–75% reduction of the corresponding mRNA in comparison with cells transduced with the scrambled-shRNA (Fig 5A and B). Knockdown of COUP-TFI and COUP-TFII protein following shRNA transduction was confirmed by Western blot (Supplementary Fig S4E). hESC-atrial CMs transduced with scrambled-shRNA or *COUP-TF*-shRNAs maintained their cardiac phenotype. Knockdown of *COUP-TFI* or *COUP-TFII* in hESC-atrial CMs did not affect GFP or *cTNT* expression (Supplementary Fig S5A and B). Furthermore, shRNA-targeted knockdown of *COUP-TFI* did not affect the expression of *COUP-TFII* and vice versa (Fig 5A and B).

However, knockdown of *COUP-TFI* or *COUP-TFII* in hESC-atrial CMs led to significant decrease in the expression of *KCNA5* (Fig 5C and D). Similarly, knockdown of *COUP-TFII* decreased the expression of ion channel genes *KCNJ3* and *KCNJ5* (Fig 5D). Although there was a small reduction in the expression of *KCNJ3* and *KCNJ5* in hESC-atrial CMs with decreased COUP-TFI expression (Fig 5C), it did not reach significance.

To test whether the atrial-enriched ion channel genes *KCNA5* and *KCNJ3* are direct targets of COUP-TFs, we performed ChIP-qPCR assays using day 30 hESC-atrial CMs. COUP-TF genes regulate transcription by interacting with direct repeats (DRs) of hormone responsive elements with various spacings, but show highest affinity to DR sequences separated by 1 nucleotide (DR1). Bioinformatic analysis of promoter regions of human *KCNA5* and *KCNJ3* by Genomatix-MatInspector revealed potential binding sites of the

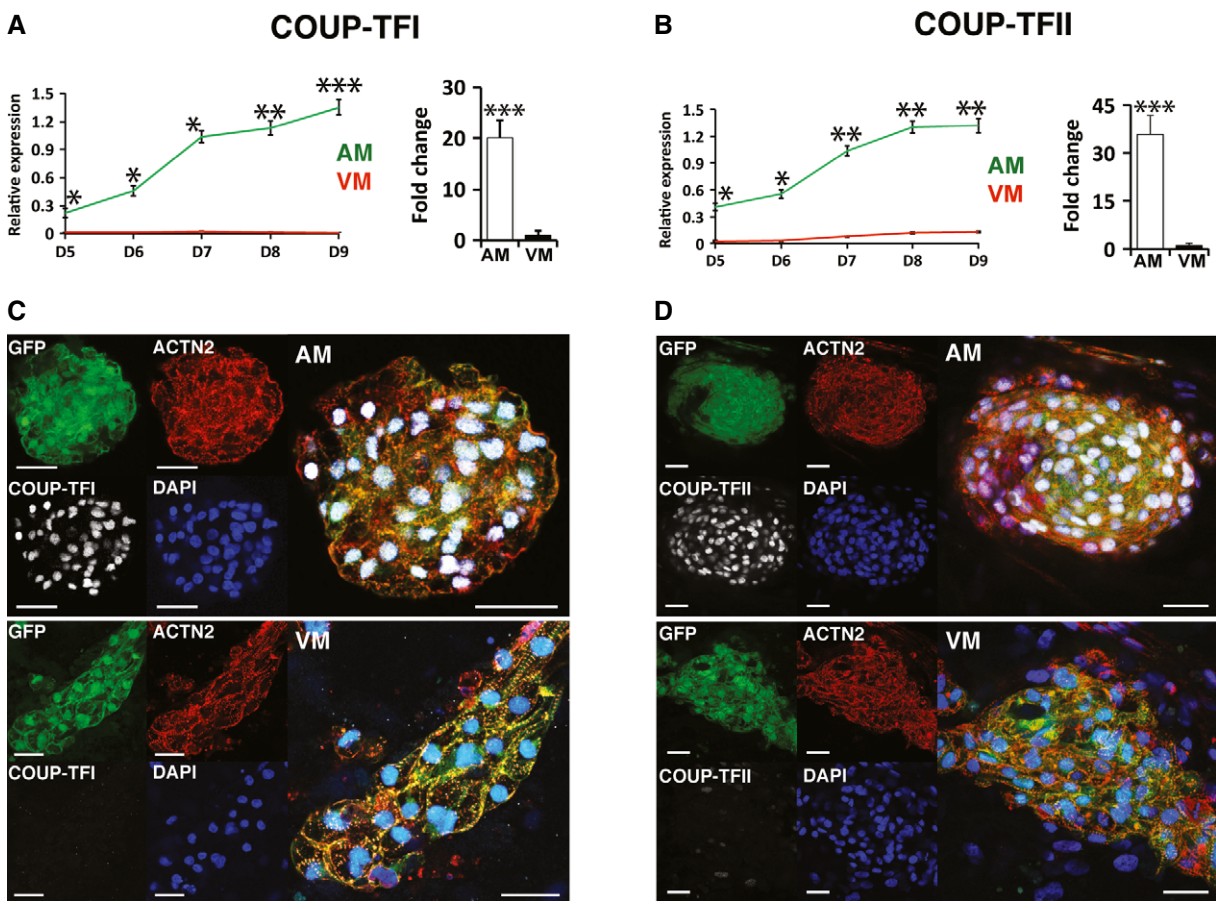

**Figure 4.  Retinoic acid induces COUP-TFI and COUP-TFII during atrial differentiation.**

A, B   Line plot illustrating relative mRNA levels of (A) *COUP-TFI* and (B) *COUP-TFII* in VM and AM differentiations from day 5 through day 9 (left) and in GFP⁺ CMs at day 31 (right); *n* = 3.

C, D   COUP-TFI (C) and COUP-TFII (D) immunofluorescence at day 31 in AM (top) and VM (bottom). Scale bars: 40 μm.

Data information: Data are presented as mean ± SEM. In (A, B), \*$P < 0.05$, \*\*$P < 0.01$, \*\*\*$P < 0.001$ by unpaired *t*-test. In (A), left panel, $P = 0.03, 0.02, 0.03, 0.005$ and $0.0004$ for comparison of COUP-TFI expression between AM and VM at days 5, 6, 7, 8 and 9 of differentiation. In (A), right panel, $P = 0.0002$ for comparison of COUP-TFI expression at day 31 between AM and VM. In (B), left panel, $P = 0.02, 0.03, 0.006, 0.004$ and $0.003$ for comparison of COUP-TFII expression between AM and VM at days 5, 6, 7, 8 and 9 of differentiation. In (B), right panel, $P = 0.0001$ for comparison of COUP-TFII expression at day 31 between AM and VM. CT = control differentiation; hESC-atrial (AM); hESC-ventricular (VM).

Genomatix-defined NR2F matrix family (Supplementary Fig S5). The promoter region of *KCNA5* harbored two plausible NR2F binding sites (Fig 5E), and analysis of immunoprecipitated DNA with primers designed around site 1 confirmed binding of both COUP-TFI and COUP-TFII (Fig 5G). Promoter analysis of *KCNJ3* identified several putative NR2F binding sites (Fig 5F), and qPCR for region encompassing site 1 confirmed interaction of both COUP-TFs (Fig 5H).

Taken together, these data suggest that COUP-TFI and COUP-TFII play a pivotal role in regulating ion channel genes responsible for unique electrophysiological phenotype of human atrial cells.

## Atrial-specific currents $I_{Kur}$ and $I_{K,ACh}$ are functional in hESC-atrial CMs

The potassium ion channels $K_v1.5$ and the $K_{ir}3.1/3.4$ are more abundant in human atrial than in ventricular CMs (Wang *et al*, 1993;

Krapivinsky *et al*, 1995) and are responsible for functional differences between the two chambers. $K_v1.5$, encoded by the gene *KCNA5*, conducts the ultrarapid delayed rectifier $K^+$ current, $I_{Kur,}$ which is a major repolarizing current in the human atrium. Heteromultimers of $K^+$ channels $K_{ir}3.1/3.4$ encoded by the genes *KCNJ3* and *KCNJ5*, respectively, conduct the acetylcholine-activated current $I_{K,ACh}$ in the human atria.

We observed significantly higher mRNA expression of both *KCNA5* and *KCNJ3* in hESC-atrial CMs compared with hESC-ventricular CMs, in a manner similar to human atrial tissue (in comparison with human adult ventricular tissue) (Fig 6A).

We next assessed the current densities of $I_{Kur}$ and $I_{K,ACh}$ in hESC-ventricular and hESC-atrial CMs. $I_{Kur}$, measured as the current sensitive to 50 μmol/l 4-AP (Wang *et al*, 1993), was clearly present in hESC-atrial CMs but absent in hESC-ventricular CMs (Fig 6B). $I_{K,ACh}$, measured as the current evoked by the muscarinic agonist, CCh (10 μmol/l), was also present in hESC-atrial CMs but could not

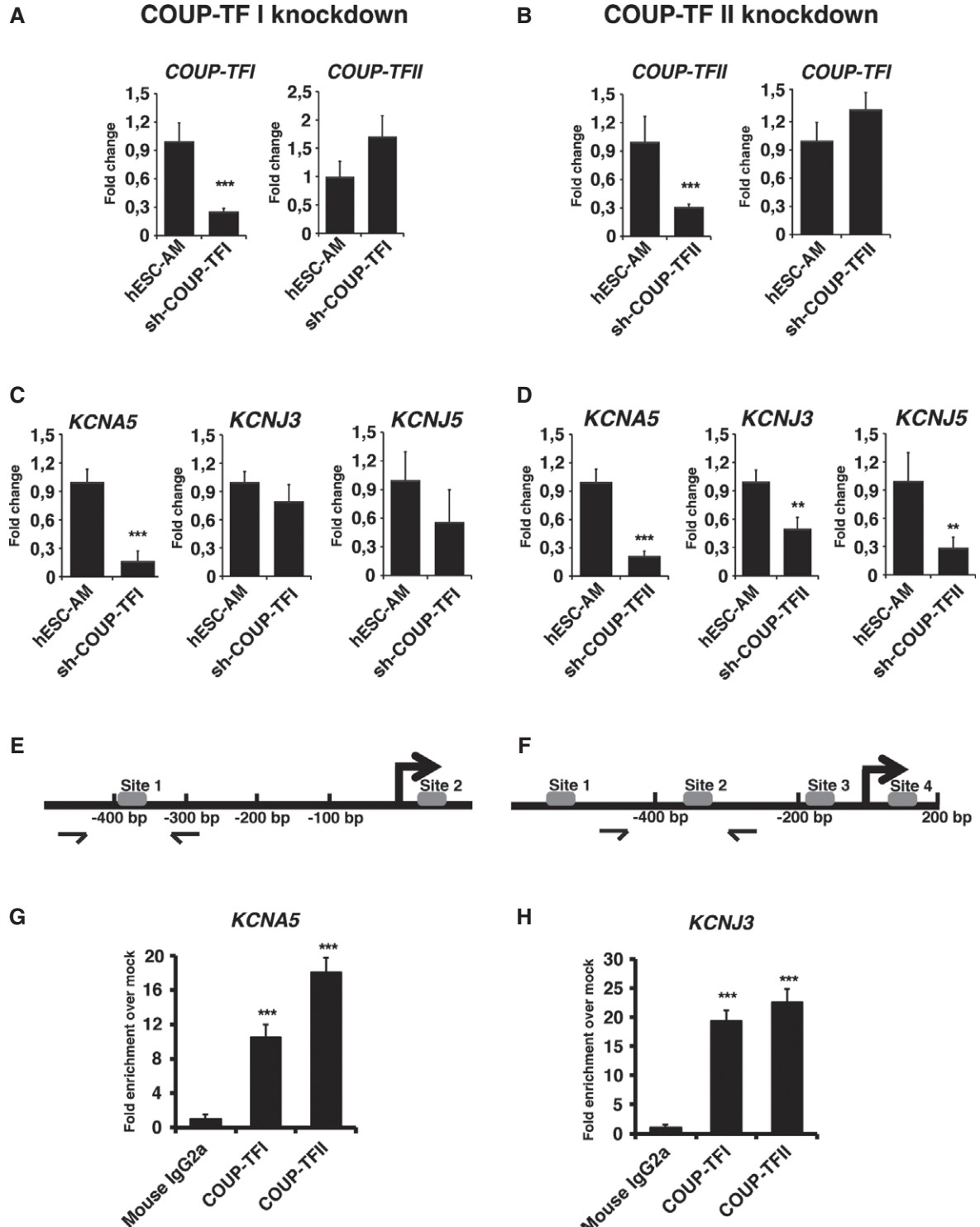

**Figure 5.  COUP-TFs regulate atrial-specific ion channel genes *KCNA5* and *KCNJ3*.**

A, B    mRNA expression of *COUP-TFI* and *COUP-TFII* following shRNA-mediated knockdown of (A) COUP-TFI or (B) COUP-TFII in hESC-atrial cardiomyocytes (AM) at day 30.

C, D    mRNA expression of ion channel genes *KCNA5*, *KCNJ3* and *KCNJ5* after knockdown of (C) COUP-TFI or (D) COUP-TFII in AM at day 30.

E, F    Schematic of NR2F binding sites in (E) *KCNA5* and (F) *KCNJ3* promoters.

G, H    ChIP-qPCR analysis at day 30 shows enriched binding of COUP-TFI and COUP-TFII to the promoter region of (G) *KCNA5* and (H) *KCNJ3*, compared to IgG in AM.

Data information: Data are presented as mean ± SEM. *$P < 0.05$, **$P < 0.01$, ***$P < 0.001$ by unpaired *t*-test. In (A), $P = 0.00004$. In (B), $P = 0.0001$. In (C), $P = 0.0001$. In (D), $P = 0.000004$, 0.006 and 0.001, respectively. In (G), $P = 0.0004$ for COUP-TFI and $P = 0.0002$ for COUP-TFII. In (H), $P = 0.0003$ for COUP-TFI and $P = 0.0001$ for COUP-TFII.

    

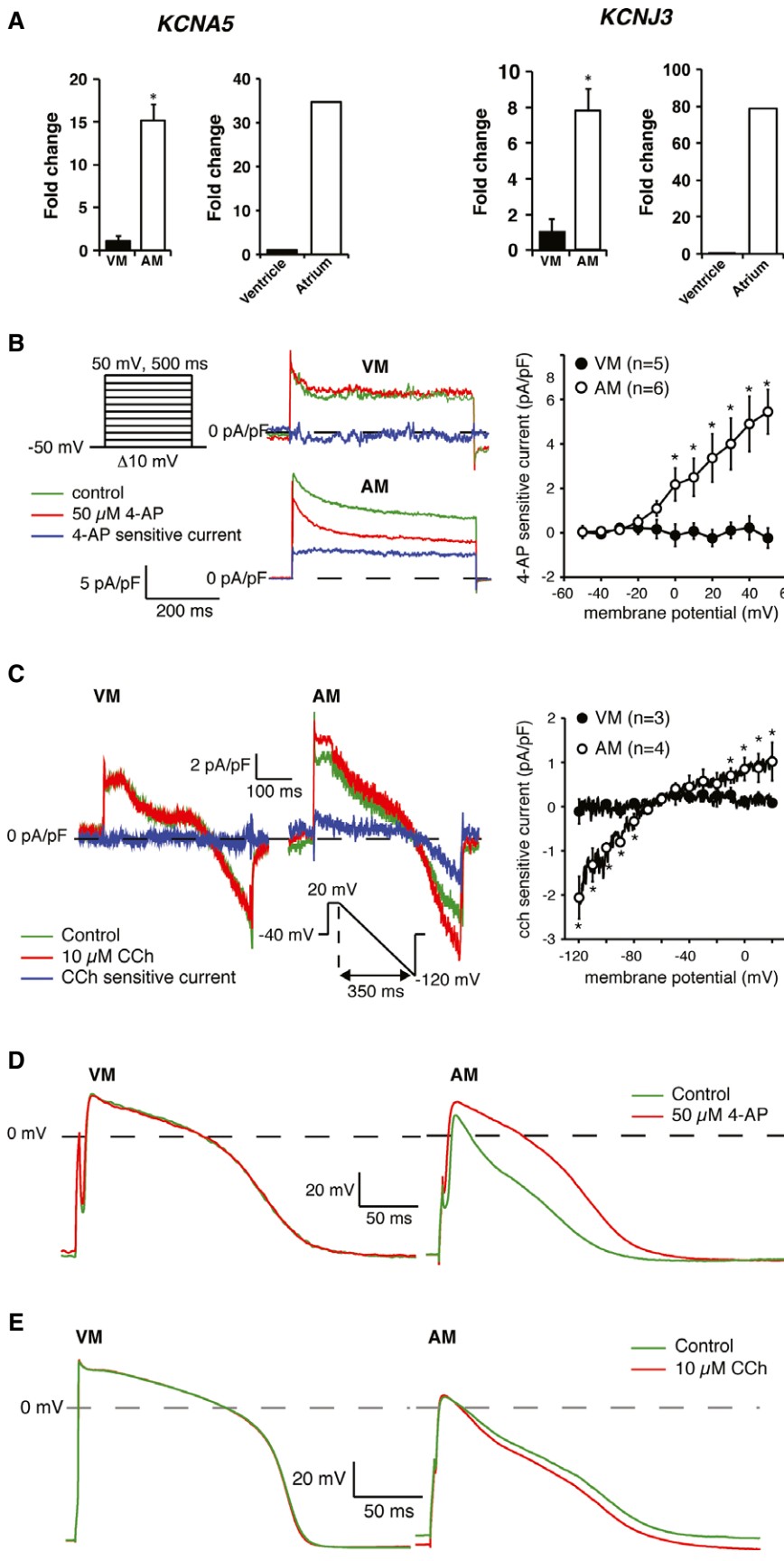

**Figure 6.**

**Figure 6.   Characterization of $I_{Kur}$ and $I_{K,ACh}$ in hESC-ventricular and hESC-atrial CMs.**

A       Expression of *KCNA5* (left) and *KCNJ3* (right) in GFP$^+$ pools of VM and AM CMs at day 31, as well as in ventricles and atria of human heart.

B, C    Typical examples (left) and current–voltage relationships (right) of (B) $I_{Kur}$ and (C) $I_{K,ACh}$ in VM and AM CMs.

D, E    Representative APs of VM and AM at 1 Hz in response to (D) $I_{Kur}$ block by 4-AP and (E) $I_{K,ACh}$ activation by CCh. AP parameters are shown in Supplementary Table S5.

Data information: Data are presented as mean ± SEM. In (A), *$P < 0.05$ by unpaired *t*-test. In (B, C), *$P < 0.05$ by two-way repeated measures ANOVA followed by pairwise comparison using the Student–Newman–Keuls test for (B) and Mann–Whitney rank-sum test for (C). In (B), $P = 0.778, 0.350, 0.03, 0.02, 0.002, 0.001, < 0.001$ and $< 0.001$, respectively, for comparison between VM and AM within membrane potentials of $-20, -10, 0, 10, 20, 30, 40$ and $50$ mV. In (C), $P = 0.01, 0.01, 0.01, 0.01, 0.03, 0.397, 0.397, 0.397, 0.671, 0.207, 0.207, 0.09, 0.01, 0.039$ and $0.015$, respectively, for comparison between VM and AM within membrane potentials of $-120, -110, -100, -90, -80, -70, -60, -50, -40, -30, -20, -10, 0, 10$ and $20$ mV. CMs = cardiomyocytes; hESC-atrial (AM) and hESC-ventricular (VM) CMs; $I_{K,ACh}$ = acetylcholine-activated potassium current; $I_{Kur}$ = potassium ultra-rapid delayed rectifier current. 4-AP = 4-aminopyridine; CCh = carbachol.

be detected in hESC-ventricular CMs (Fig 6C). Thus, hESC-atrial CMs have substantially higher $I_{Kur}$ and $I_{K,ACh}$ densities, consistent with the greater mRNA expression of *KCNA5* and *KCNJ3*.

Lastly, we evaluated the contribution of $I_{Kur}$ and $I_{K,ACh}$ in the APs of hESC-atrial and hESC-ventricular CMs. Blocking $I_{Kur}$ by 4-AP (50 μmol/l) reduced phase-1 repolarization resulting in AP prolongation and an increase in APA$_{plat}$ in hESC-atrial but not hESC-ventricular CMs (Supplementary Fig S6D and Supplementary Table S5). These effects of $I_{Kur}$ block observed in hESC-atrial CMs confirmed its functional presence and are consistent with the effects reported in freshly isolated human atrial CMs (Wang *et al*, 1993). On the other hand, activation of $I_{K,ACh}$ by CCh resulted in hyperpolarization of the RMP in hESC-atrial CMs but not in hESC-ventricular CMs (Fig 6E and Supplementary Table S5). The effects of $I_{K,ACh}$ activation on the AP of hESC-atrial CMs are consistent with findings in isolated human atrial myocytes (Koumi *et al*, 1994).

These results suggest that hESC-atrial CMs derived from RA-treated differentiations possess functional $I_{Kur}$ and $I_{K,ACh}$ currents and might therefore be a suitable model for testing drug responses of pharmacological compounds selective for atrial cells.

### Effects of vernakalant on APs of hESC-atrial and hESC-ventricular CMs

To validate hESC-atrial CMs as a preclinical model for screening the selectivity of ion channel blockers, we tested the effects of antiarrhythmic agent, vernakalant (Wettwer *et al*, 2013). Intravenous form of this drug has recently been approved by the European Medicines Agency for cardioversion of recent-onset AF (Savelieva *et al*, 2014). In order to study the effects of this compound on the APs of hESC-atrial and hESC-ventricular CMs, 30 μmol/l of the compound (Wettwer *et al*, 2013) was administered while the cells were paced at various frequencies (1–4 Hz) and compared with pre-drug controls.

Figure 7A shows typical APs of hESC-atrial and hESC-ventricular CMs at 1 Hz in the absence and presence of vernakalant. In hESC-atrial CMs, at a frequency of 1 Hz, vernakalant significantly reduced dV/dt$_{max}$ (Fig 7A, inset) and increased APA$_{max}$ ($> 2.5$ mV) and APA$_{plat}$ ($> 20\%$) resulting in prolongation of early as well as late repolarization (APD$_{20}$: $> 7.5$ ms; APD$_{50}$: $> 15$ ms and APD$_{90}$: $> 22$ ms) in hESC-atrial CMs. These effects on APA$_{plat}$ and dV/dt were also observed at higher stimulation frequencies (Fig 7B and C). In hESC-ventricular CMs, vernakalant reduced dV/dt$_{max}$ but without affecting other AP parameters at 1 Hz (Fig 7A and Supplementary Table S6). Interestingly, vernakalant depressed dV/dt$_{max}$ in a frequency-dependent manner in both hESC-atrial and

hESC-ventricular CMs by a similar amount (Fig 7C). The AP changes in response to vernakalant were nearly reversible upon washout.

The effects of 30 μmol/l vernakalant on dV/dt$_{max}$, APA and APD$_{20}$ of hESC-atrial CMs were consistent with the results observed in human atrial trabeculae in sinus rhythm (SR) (Wettwer *et al*, 2013).

### Effects of XEN-D0101 on APs of hESC-atrial and hESC-ventricular CMs

Drugs developed to target K$_v$1.5 channels would ideally offer atrial selectivity and have no proarrhythmic effect, since $I_{Kur}$ conducted by these channels is absent in the ventricles. To determine the response of hESC-atrial CMs to blockers that act on repolarizing potassium currents expressed preferentially in atrial CMs, we tested the effect of a selective K$_v$1.5 blocker, XEN-D0101 (Ford *et al*, 2013). Figure 8A shows typical APs of hESC-atrial and hESC-ventricular CMs at 1 Hz in the absence and presence of 3 μmol/l XEN-D0101. Treatment with XEN-D0101 caused robust elevation of APA$_{plat}$ ($> 26$ mV) as well significant prolongation of APD$_{20}$ ($> 30$ ms), APD$_{50}$ ($> 35$ ms) and APD$_{90}$ ($> 23$ ms) in hESC-atrial cells, but the compound did not significantly alter any AP parameter in hESC-ventricular CMs (Fig 8A and Supplementary Table S7). AP changes caused by XEN-D0101 were reversible upon washout. The effect of XEN-D0101 on APD$_{20}$ and APD$_{50}$ of hESC-atrial CMs is consistent with the effects observed in native human atrial trabeculae in SR (Ford *et al*, 2013). On the contrary, XEN-D0101 significantly altered APA ($> 10$ mV) and dV/dt$_{max}$ ($> 8$ V/s) in hESC-atrial CMs compared with atrial trabeculae in SR. Intriguingly, XEN-D0101 prolonged APD$_{90}$ in hESC-atrial CMs as well as in human atrial trabeculae (Ford *et al*, 2013) in AF while a reduction was observed in SR.

### Effects of XEN-R0703 on APs of hESC-atrial and hESC-ventricular CMs

Enhanced parasympathetic tone and constitutive activation of $I_{K,ACh}$ are believed to be contributing factors to both paroxysmal AF (clinically termed 'vagal AF') and chronic AF in man (Dobrev *et al*, 2005). Thus, antiarrhythmic drugs targeting the K$_{ir}$3.1/3.4 channels are a promising therapeutic option for AF termination and the maintenance of SR. To determine the presence of $I_{K,ACh}$ in hESC-atrial and hESC-ventricular CMs, we tested XEN-R0703, a novel selective $I_{K,ACh}$-blocking antiarrhythmic drug. The ion channel pharmacology of XEN-R0703 was investigated in HEK293 cells or CHO cells expressing the channel of interest (Supplementary Table S8 and

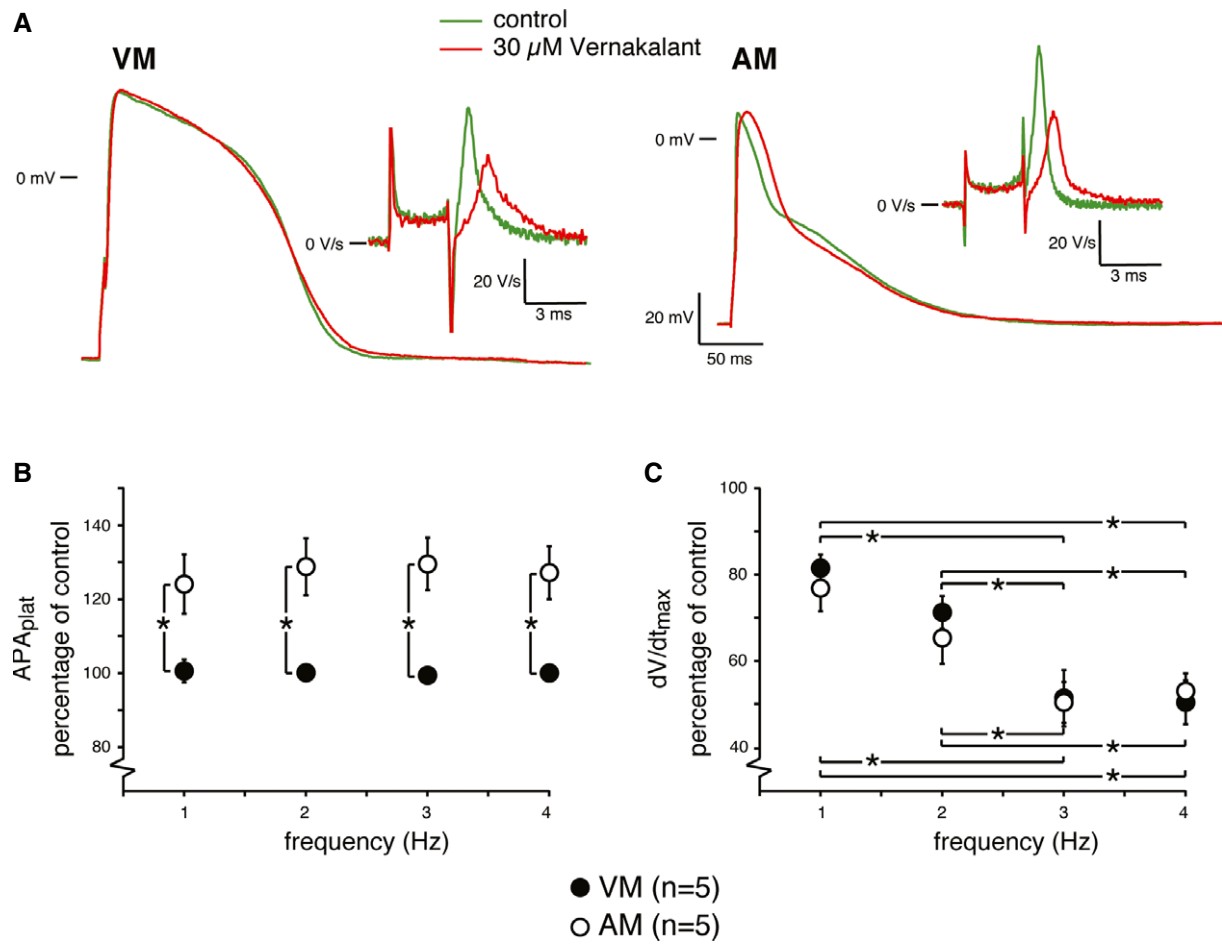

**Figure 7. Effects of vernakalant on APs of hESC-ventricular and hESC-atrial CMs.**

A    Representative APs at 1 Hz of VM and AM CMs in response to vernakalant. Inset shows dV/dt$_{max}$.

B, C  Average APA$_{plat}$ (B) and dV/dt$_{max}$ (C)in the absence and presence of vernakalant at 1–4 Hz. AP parameters are shown in Supplementary Table S6. Abbreviations as in Figs 3 and 6.

Data information: Data are presented as mean ± SEM. *$P < 0.05$ by Mann–Whitney rank-sum test for (B). $P = 0.01$, 0.008, 0.008 and 0.007, respectively, for comparison of APA$_{plat}$ between VM and AM groups at frequencies of 1.0, 2.0, 3.0 and 4.0 Hz. Two-way repeated measures ANOVA followed by pairwise comparison using the Student–Newman–Keuls test for (C). $P = 0.686$ between VM and AM groups and hence not statistically significant. For VM, $P = 0.06$ for 1 versus 2 Hz; $P < 0.001$ for 1 versus 3 Hz; $P < 0.001$ for 1 versus 4 Hz; $P = 0.001$ for 2 versus 3 Hz; $P = 0.002$ for 2 versus 4 Hz; and $P = 0.857$ for 3 versus 4 Hz. For AM, $P = 0.03$ for 1 versus 2 Hz; $P < 0.001$ for 1 versus 3 Hz; $P < 0.001$ for 1 versus 4 Hz; $P = 0.02$ for 2 versus 3 Hz; $P = 0.02$ for 2 versus 4 Hz; and $P = 0.621$ for 3 versus 4 Hz.

Supplementary Fig S7). XEN-R0703 potently inhibited recombinant K$_{ir}$3.1/3.4 (IC$_{50}$ = 57 nM, n$_H$ = 0.52 ± 0.1) and had nominal effect on other cardiac channels displaying 100-fold selectivity over hERG (IC$_{50}$  5.6 µM, n$_H$ = 0.99 ± 0.2) and > 300-fold selectivity over Na$_v$1.5, Ca$_v$1.2 and K$_{ir}$2.1 (IC$_{50}$ ≫ 10 µM for each) (Supplementary Fig S7 and Supplementary Table S8). The effect of XEN-R0703 on native human I$_{K,ACh}$ was confirmed using primary human atrial CMs. Of 300 nM XEN-R0703 inhibited the CCh-activated native human I$_{K,ACh}$ by 81 ± 10% (Supplementary Fig S8).

To study the effect of 1 µmol/l XEN-R0703 on APs of hESC-atrial and hESC-ventricular CMs, I$_{K,ACh}$ current was first activated with 10 µmol/l CCh as shown in Fig 6, followed by the addition of XEN-R0703. Figure 8B shows typical APs of hESC-atrial and hESC-ventricular CMs at 1 Hz in the absence and presence of XEN-R0703 as well as in the continuous presence of CCh. XEN-R0703 reversibly depolarized the RMP and restored the AP shortening caused by

CCh in hESC-atrial CMs (Supplementary Table S9). In contrast to hESC-atrial CMs, XEN-R0703 did not affect any AP parameter in hESC-ventricular CMs, consistent with the absence of I$_{K,ACh}$ in these cells. These results also demonstrate that XEN-R0703 does not affect other membrane currents present in hESC-ventricular CMs, consistent with the effects of the drug observed in HEK-293 cells expressing various ion channels (Supplementary Table S8 and Supplementary Fig S7).

In order to confirm the selectivity of XEN-R0703 predicted by the effect on hESC-atrial CMs in vitro, the effect of XEN-R0703 was studied in an in vivo RAP dog model of persistent AF. XEN-R0703 increased right atrial effective refractory period (AERP) in dog by 10, 17 and 28% at 1, 3 and 10 mg/kg without affecting the Van de Water's QTc interval (Fig 8C). Furthermore, as depicted in Fig 8D, XEN-R0703 reduced AF inducibility from 76% in vehicle to 43 and 11% following the administration of 3 and 10 mg/kg.

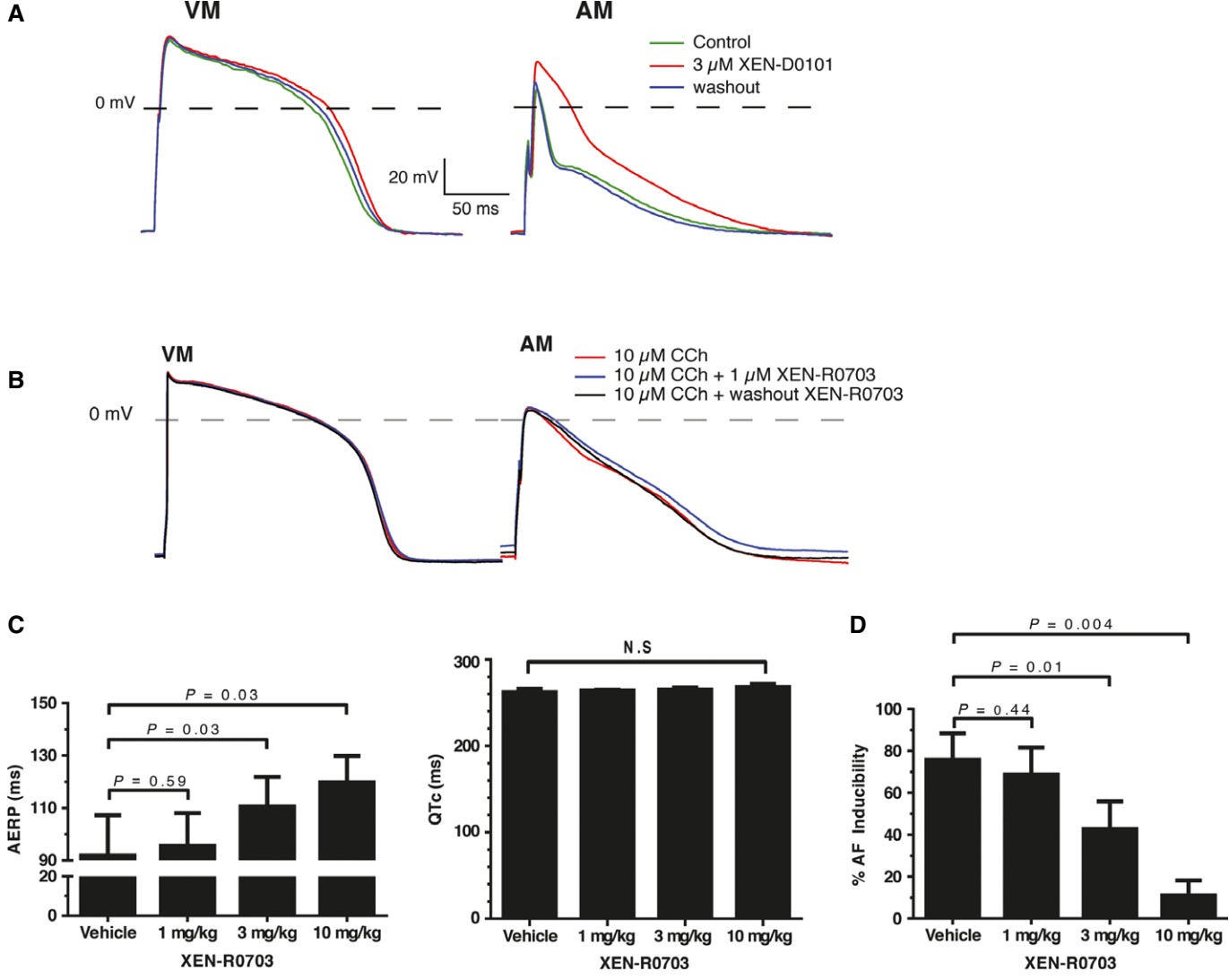

**Figure 8.  Effects of XEN-D0101 and XEN-R0703 on APs of hESC-ventricular and hESC-atrial CMs.**

A     Representative APs of VM and AM CMs in the absence, presence and following washout of 3 μmol/l XEN-D0101. AP parameters are shown in Supplementary Table S7.

B     Representative APs (1 Hz) of VM and AM in the CCh, to activate $I_{K,ACh}$ and subsequent addition of XEN-R0703. AP parameters are shown in Supplementary Table S9.

C, D   Experiments performed in RAP conscious dogs in the presence of vehicle or following 1, 3 and 10 mg/kg XEN-R0703 show (C) mean right AERP values (left), mean Van de Water's QTc (right) and (D) AF inducibility plotted as a function of dose.

Data information: For RAP dog experiments, *n* = 5; statistical significance tested with paired *t*-test. Data are presented as mean ± SEM. AERP = atrial effective refractory period; AF = atrial fibrillation; RAP = rapid atrial pacing; N.S. = not significant. Other abbreviations as in Figs 3, 6 and 7.

## Discussion

Despite the remarkable improvement in efficiency and robustness of protocols for cardiac differentiation of hPSCs, the resulting CM population is usually a heterogeneous pool of atrial-, ventricular- and nodal-like cells (Mummery *et al*, 2012). Native atrial and ventricular CMs exhibit distinct molecular and functional profiles essential for their diverse physiological roles in the heart, and hPSC-CM cultures enriched in these subtypes would have significant added value in drug response assays.

In the study described here, we directed hESCs toward atrial-like CMs by exogenous addition of RA during CM differentiation.

RA signaling is crucial for atrial chamber development *in vivo* (Niederreither *et al*, 2001; Hochgreb *et al*, 2003), and its activation has previously been shown to steer differentiation of mouse and human ESCs toward atrial-like CMs (Gassanov *et al*, 2008; Zhang *et al*, 2011). However, little is known about the molecular mediators that govern ion channel repertoire of RA-driven atrial-like CMs and their suitability as a model system for preclinical drug screenings.

Gene expression profiling of the resulting CMs, exposed to RA during differentiation, indicated an upregulation of atrial transcripts such as *COUP-TFII*, *SLN*, *NPPA* and *PITX2* along with a downregulation of ventricular transcripts such as *HAND1*, *HEY2*, *IRX4* and

*MYL2*. *COUP-TFII* is expressed in atrial chambers of the heart and has been reported to determine atrial identity in mice (Wu *et al*, 2013). *SLN* is also atrial specific and is an integral part of the sarcoplasmic reticulum calcium complex (Minamisawa *et al*, 2003). *NPPA* is expressed in both atrial and ventricular chambers during development and becomes progressively restricted to the atria shortly after birth (Chuva de Sousa Lopes *et al*, 2006). Cardiac left–right determinant *PITX2* is expressed in the left atrium, and its insufficiency has been linked to atrial arrhythmogenesis (Kirchhof *et al*, 2011). Myosin light chain gene, *MYL2,* iroquois homeobox transcription factor, *IRX4,* and basic helix-loop-helix transcription factors, *HAND1* and *HEY2,* are expressed in the developing ventricles of the heart and have critical roles in ventricular chamber development and function (Moorman & Christoffels, 2003).

Moreover, the global gene profile of hESC-atrial CMs showed a higher overlap with that of human fetal atria, and hESC-ventricular CMs showed an increased overlap with that of human fetal ventricles. About 30% of the genes in GFP$^+$ cells of hESC-atrial or hESC-ventricular CMs overlapped with that of chamber-specific genes in the fetal heart. Taking into account that we used atria and ventricles from a 15-week-old human heart, consisting of cardiomyocytes, endothelial cells, smooth muscle cells, fibroblasts and other cardiac cells, a 30% overlap can be considered as a strong correlation. An additional limiting factor is the stage of heart used for microarray analysis. We used a second trimester human fetal heart for comparison to hESC-derived CMs while a previous study has noted that ESC-derived CMs resemble that of embryonic heart tube (Fijnvandraat *et al*, 2003).

Multiple mechanisms such as transcriptional regulation or programmed cell death leading to selective survival might contribute to subtype specification of atrial or ventricular CMs. Earlier studies in amniotes had suggested a possibility that RA is required for the formation of atrial cardiomyocytes and that in its absence, cardiac precursors differentiate to ventricular cells (Hochgreb *et al*, 2003; Simoes-Costa *et al*, 2005). However, a recent study in zebrafish has shown that RA signaling acts via different mechanisms to limit both atrial and ventricular cell numbers but not at the expense of each other (Waxman *et al*, 2008). In the current study, we did not observe any unusual apoptosis in RA-treated embryos compared to controls during hPSC differentiation. Moreover, we observed rapid and robust induction of COUP-TFs in response to RA and thus proposed a central role for these transcription factors in RA-driven atrial differentiation. COUP-TFI and COUP-TFII belong to the steroid receptor super family of genes and display overlapping yet distinct patterns of expression in all the three germ layers in mouse (Pereira *et al*, 2000). Both these COUP-TF genes are induced by retinoids *in vivo* in zebrafish brain (Jonk *et al*, 1994). In addition, determination of the crystal structure of COUP-TFII led to its identification as a RA-activated receptor (Kruse *et al*, 2008). Interestingly, deletion of either *RALDH2* or *COUP-TFII* in the mouse results in severe abnormalities of the atria and sinus venosus, implicating *COUP-TFII* as a possible downstream effector of RA-driven posterior chamber specification.

While a role for COUP-TFII in the heart and vasculature has been identified, little is known about the function of COUP-TFI in the heart. COUP-TFI was found to be expressed in whole heart protein lysates obtained from embryonic and neonatal mouse hearts and has been proposed to antagonize the activation of calreticulin

promoter by NKX2.5 (Guo *et al*, 2001). The localization of COUP-TFI in the human heart and its role in cardiac lineage specification have been unknown to date. We showed here that COUP-TFI is induced by RA along with COUP-TFII during atrial differentiation *in vitro* and was also expressed specifically in the atrial chambers of the human heart. It is worth mentioning that COUP-TFI was observed only in the atria of the human fetal heart while expression of COUP-TFII spanned a broader region (endothelium, smooth muscle cells—data not shown) than just the atria, suggesting non-redundant functions of these genes in the human heart.

Based on our results as well as evidence from previous studies pointing to an integral role for COUP-TFs in the retinoid network and cell-fate determination, we investigated the functional significance of robust expression of these genes in hESC-atrial CMs. shRNA-mediated knockdown as well as ChIP experiments demonstrated that COUP-TFs regulate the atrial-selective ion channel gene, *KCNA5*. These experiments also established that *KCNJ3* and *KCNJ5* are regulated by COUP-TFII. Interestingly, although COUP-TFI showed strong interaction with the *KCNJ3* promoter, knockdown of COUP-TFI itself did not result in a decrease in the expression of *KCNJ3*. This suggests that COUP-TFI might be dispensable for the expression of *KCNJ3* and *KCNJ5* in atrial CMs. Future studies aimed at dissecting the roles of COUP-TFI and COUP-TFII during atrial differentiation are required to understand whether these genes act in synergy or possess functions independent and specific to one another. It also remains to be tested whether loss of COUP-TFI can be rescued by COUP-TFII and *vice versa*. However, since myocardial ablation of COUP-TFII in the mouse results in a severe phenotype (Wu *et al*, 2013), it seems unlikely that COUP-TFI can account for loss of COUP-TFII. Nonetheless, it is likely that deletion of both COUP-TFI and COUP-TFII might result in a more severe phenotype.

Ion channels Kv1.5, Kir3.1 and Kir3.4 encoded by *KCNA5, KCNJ3* and *KCNJ5,* respectively, conduct the potassium currents $I_{Kur}$ and $I_{K,ACh,}$ which are major determinants of electrophysiological differences between atrial and ventricular CMs (Schram *et al*, 2002; Ravens *et al*, 2013) in humans. Although mechanisms controlling ion channel expression are far more complex than transcriptional regulation alone, these findings identify a potential regulatory mechanism of *KCNA5* and *KCNJ3* in human atrial myocytes. Previous work by others has shown that COUP-TFs regulate the expression of $Na^+/H^+$exchanger (NHE) in differentiating P19 cells (Fernandez-Rachubinski & Fliegel, 2001). The ability of these transcription factors to regulate ion channels in diseased and non-diseased human heart warrants further investigation to better understand their role in pathophysiological states.

In addition to identifying a central role for COUP-TFs in the transcriptional regulation of *KCNA5* and *KCNJ3*, we also reported the functional presence of potassium currents encoded by these atrial-specific ion channel genes, in hESC-atrial but not hESC-ventricular CMs. These findings prompted us to investigate whether hESC-atrial CMs would be a suitable model for preclinical testing of pharmacological compounds currently being developed for AF. Drug responses of hESC-atrial CMs following treatment with multiple ion channel blocker, vernakalant, and selective $K_v1.5$ blocker, XEN-D0101, recapitulated the effects observed on early repolarization in human right atrial trabeculae in SR (Ford *et al*, 2013; Wettwer *et al*, 2013). Furthermore, we observed effects of these compounds on other AP parameters such as APD$_{90}$ in

hESC-atrial CMs that differ from those previously reported in human atrial trabeculae in SR. This may be due to the use of dialyzed (whole-cell patch-clamp) single cells in this study, whereas previous studies used multicellular preparations, in which cells were non-dialyzed (sharp microelectrode) and electrically coupled. Additionally, we tested a novel $K_{ir}3.1/3.4$ blocker, XEN-R0703, in an *in vivo* RAP dog model. Dog is regarded as the one of the most predictive animal species of human cardiovascular toxicity and is routinely used in the pharmaceutical industry to assess cardiac safety and the risk of drug-induced ventricular arrhythmias (Olson *et al*, 2000). In the dog, XEN-R0703 resulted in a dose-dependent increase of AERP without affecting the QTc interval. These results suggest lack of effect of XEN-R0703 on the ventricles and confirm its atrial selectivity as predicted in hESC-atrial CMs.

hPSC-derived CMs are spontaneously active and exhibit depolarized RMP. Evidence suggests that they are developmentally immature, resembling human fetal CMs rather than their adult counterparts (Beqqali *et al*, 2006). In our study, RMP did not differ between hESC-atrial CMs and hESC-ventricular CMs indicating similar level of maturity in both the groups. Nonetheless, hESC-atrial CMs respond to atrial-selective ion channel blockers demonstrating that an immature electrical phenotype does not preclude their use in preclinical drug screening and pharmacology. Our findings have important implications for integrating hPSC-derived atrial CMs into high-throughput screenings for selection and validation of lead compounds during early stages of drug discovery.

In conclusion, we addressed the void for a humanized preclinical screening platform in the pharmaceutical industry for evaluating selectivity of novel ion channel blockers for AF. We showed that hESC-atrial CMs respond to atrial-selective compounds in a manner similar to isolated human atrial CMs, thus demonstrating the potential of this tool as a robust model for preclinical atrial-selective pharmacology.

# Materials and Methods

For more detailed Materials and Methods, please see Supplementary Information.

## hESC culture and differentiation to CMs

A transgenic *NKX2-5-eGFP/w* hESC line that faithfully reports endogenous NKX2.5 expression by GFP was described previously (Elliott *et al*, 2011). Undifferentiated hESCs were maintained on irradiated mouse embryonic fibroblasts, and cardiac differentiation was induced using a spin EB protocol. Briefly, hESCs were harvested and resuspended on day 0 in BPEL medium (Ng *et al*, 2008) containing 20–30 ng/ml hActivin-A (R&D Systems), 20–30 ng/ml bone morphogenetic protein 4 (R&D Systems), 40 ng/ml stem cell factor (Stem Cell Technologies), 30 ng/ml vascular endothelial growth factor (R&D Systems) and 1.5 μmol/l CHIR 99021 (Axon Medchem). EBs were refreshed on day 3 with BPEL and then transferred to gelatin-coated dishes on day 7.

To induce atrial specification in hESCs, cardiac differentiation was initiated as described above and 1 μmol/l all-*trans* retinoic acid (RA) (Sigma) was added on day 4 of differentiation. Cells were refreshed with BPEL on day 7 of differentiation.

## Cellular electrophysiology

### Cell preparation, data acquisition and analysis

Spin EBs resulting from control and RA-treated differentiations were dissociated at day 17 to single cells using TrypLE™ Select (Life Technologies) and plated on gelatin-coated coverslips. Electrophysiological measurements were performed 10–14 days after dissociation from intrinsically quiescent single $GFP^+$ CMs that were able to contract upon field stimulation.

APs and membrane currents from hESC-CMs were recorded with the amphotericin-B-perforated patch-clamp technique at $36 \pm 0.2°C$ using an Axopatch 200B amplifier (Molecular Devices, Sunnyvale, CA, USA). Cells were superfused with Tyrode's solution containing (in mmol/l) NaCl 140, KCl 5.4, $CaCl_2$ 1.8, $MgCl_2$ 1.0, glucose 5.5, and HEPES 5.0; pH was adjusted to 7.4 with NaOH. Pipettes (borosilicate glass; resistance 2–3 MΩ) were heat polished and filled with solution containing (in mmol/l) K-gluconate 125, KCl 20, NaCl 5, amphotericin-B 0.22 and HEPES 10; pH was adjusted to 7.2 with KOH. AP measurements were low-pass-filtered (cutoff frequency 10 kHz) and digitized at 40 kHz; membrane currents were measured at 1 and 4 kHz, respectively. Capacitance and series resistance were compensated by $\geq 80\%$, and APs were corrected for the calculated liquid junction potential (Barry & Lynch, 1991). Voltage control, data acquisition and analysis were accomplished using custom software. Cell membrane capacitance ($C_m$) was estimated by dividing the time constant of the decay of the capacitive transient in response to 5 mV hyperpolarizing voltage clamp steps from −40 mV by the series resistance.

### Current clamp experiments

APs were elicited at 0.5 to 4 Hz by 3 ms, ~1.2× threshold current pulses through the patch pipette. APs were characterized, as depicted in Fig 2A, by resting membrane potential (RMP), maximum upstroke velocity ($dV/dt_{max}$), maximum AP amplitude ($APA_{max}$), AP plateau amplitude ($APA_{plat}$, defined as the potential difference between RMP and potential at 20 ms after the upstroke), and the duration at 20, 50 and 90% repolarization ($APD_{20}$, $APD_{50}$, and $APD_{90}$, respectively). Parameter values obtained from 10 consecutive APs were averaged.

### Voltage clamp experiments

The ultrarapid delayed rectifier $K^+$ current ($I_{Kur}$) and the acetylcholine-activated $K^+$ current ($I_{K,ACh}$) were measured as the current sensitive to 50 μM 4-aminopyridine (4-AP) or 10 μM carbachol (CCh), respectively, in the presence of 10 μM nifedipine to block the L-type $Ca^{2+}$ current. Voltage clamp protocols are shown in the corresponding figures and have been described previously (Choisy *et al*, 2012; Wang *et al*, 1993). Current density was calculated by dividing current amplitude by $C_m$.

## Statistics

qPCR, electrophysiology, pharmacology, ChIP and knockdown experiments were performed on cardiomyocytes resulting from three independent control and RA-treated differentiations.

Statistical analysis was carried out with SigmaStat 3.5 software. Normality and equal variance assumptions were tested with the Kolmogorov–Smirnov and the Levene median test, respectively.

Groups were compared with unpaired *t*-test or with Mann–Whitney rank-sum test (in case of a failed normality and/or equal variance test). Two-way repeated measures (RM) ANOVA followed by the Student–Newman–Keuls *post hoc* test was used by comparing groups in the frequency and I–V relationships. In case of a failed normality and/or equal variance test, data were tested with the Mann–Whitney rank-sum test per frequency or voltage. Paired *t*-tests were used to compare drug effects within a group of cells. Data obtained at a series of frequencies within a group were compared with one-way RM ANOVA. Groups were compared using paired/unpaired *t*-test or two-way repeated measures ANOVA followed by pairwise comparison using the Student–Newman–Keuls test. $P < 0.05$ defines statistical significance. Data are presented as mean ± SEM.

### Ethics statement

Studies on hESCs were performed in the Netherlands, and their use was approved by the medical ethical committee of Leiden University Medical Center (LUMC). Collection and use of human fetal material for research was also approved by the medical ethical committee of LUMC (protocol 08.087). Specimens of human atrial appendage were obtained from patients undergoing a range of cardiac surgical procedures with written informed consent and conformed to the principles outlined in the Declaration of Helsinki. Tissue was obtained from consenting patients (from Papworth Hospital NHS Trust, Cambridge, UK) following approval from the Local Research Ethical Approval Committee (H03/035). Animal experiments were carried out by CorDynamics, IL, USA, in compliance with the Guide for the Care and Use of Laboratory Animals (U.S.A. NIH publication No 85-23, revised 1985).

Supplementary information for this article is available online: http://embomolmed.embopress.org

### Acknowledgements

Funding for this study from the following sources is gratefully acknowledged: Netherlands Organization for Health Research and Development (ZonMw-TOP 40-00812-98-12086) to HDD and (ZonMw-MKMD-40-42600-98-036) to RP; Leiden University Medical Center (BW-plus) doctoral grant to VR; European Union (FP7-Health T2-2010-261057 'EUTRAF') to Xention Ltd. (JWF, JTM, CJ, SE-H); Netherlands Heart Foundation (NHS 2008B106) to KG; Netherlands Organization for Scientific Research (NWO-ASPASIA 016.121.365) and Interuniversity Attraction Poles Program (IUAP-07/07) to SMCdSL; and European Research Council advanced grant (STEMCARDIOVASC-323182) to CLM. The authors thank Dr. Milena Bellin for critical reading of the manuscript and CASA (Leiden and Den Haag) for the collection of human fetal material.

### Author contributions

HDD designed the study, performed hESC differentiations, characterization of cardiomyocytes, microarray data analysis, ChIP experiments and interpretation of electrophysiology data. VS performed differentiations and COUP-TF shRNA experiments. JWF and JTM performed protocol design, data analysis and interpretation of experiments in native human myocytes and recombinant ion channel screening. SEH and CJ performed data acquisition, analysis and interpretation of experiments in human myocytes and recombinant ion channel screening. KG performed gene ontology analysis. DAE provided the NKX2-5 (eGFP/w) hESC line and SMCdSL provided the fetal heart material. CLM approved the final manuscript. AOV performed protocol design, data acquisition, analysis and interpretation of electrophysiology experiments. RP supervised the study and approved the final manuscript. HDD wrote the manuscript with contributions from JTM and AOV.

### Conflict of interest

CLM and RP are co-founders and advisors of Pluriomics. JWF, JTM, SE-H and CJ are employees of Xention Ltd and hold stock/stock options in Xention Ltd. The remaining authors declare that they have no conflict of interest.

### For more information

http://www.heart.org/atrialfibrillation

https://www.lumc.nl/org/anatomie-embriologie/research/902041006062533/

https://www.lumc.nl/org/anatomie-embriologie/medewerkers/
1112140234062531

www.xention.com

## The paper explained

### Problem
Atrial fibrillation (AF) is the most common sustained arrhythmia and it imposes a huge socioeconomic burden worldwide. Existing antiarrhythmic drugs for the treatment of AF carry the risk of causing ventricular proarrhythmia or negative ionotropy. This highlights the need for developing atrial-selective drugs that promise safety and efficacy. However, current preclinical screening assays to identify atrial-specific drugs use non-cardiac cell lines or animal models, both of which have limitations in predicting the drug responses on the human heart.

### Results
We generated and characterized human embryonic stem cell-derived atrial-like cardiomyocytes (hESC-atrial CMs). We identified that COUP-TF transcription factors, induced in response to retinoic acid during atrial differentiation, regulate atrial-specific ion channel genes in hESC-atrial CMs. Furthermore, we tested the effect of Vernakalant, a recently approved drug for the treatment of AF in Europe and noted that hESC-atrial CMs elicited effects comparable to those observed in native human CMs in sinus rhythm. We also showed that hESC-atrial CMs predict atrial-selectivity of novel ion channel blockers, XEN-D0101 and XEN-R0703. Collectively, these results demonstrate that hESC-atrial CMs are a valuable model for preclinical drug screenings to identify effective atrial-selective compounds.

### Impact
We uncovered a role for COUP-TFI and COUP-TFII in hESC-atrial CMs, which warrants further investigation into the role of these transcription factors in cardiac development and disease. Moreover, generation and characterization of cardiomyocyte subtypes such as atrial, ventricular and pacemaker cells *in vitro* is essential for their application in pre-clinical and clinical testing. In this study, we addressed the impending need for a preclinical screening model resembling the physiology of a human atrial cardiomyocyte. The finding that hESC-atrial CMs are a robust model for atrial-selective pharmacology has major implications for drug discovery and development to combat AF.

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
