## [Review Process File · EMBO Molecular Medicine]

Atrial-like cardiomyocytes from human pluripotent stem cells are a robust preclinical model for assessing atrial-selective pharmacology

Harsha D. Devalla; Verena Schwach; John W. Ford; James T. Milnes; Said El-Haou; Claire Jackson; Konstantinos Gkatzis, David A. Elliott; Susana M. Chuva de Sousa Lopes; Christine L. Mummery; Arie O. Verkerk and Robert Passier

Corresponding author: Harsha Devalla, Leiden University Medical Center

Review timeline:

Submission date:	17 October 2014
Editorial Decision:	13 November 2015
Revision received:	07 December 2014
Editorial Decision:	23 December 2014
Accepted:	23 January 2015

Transaction Report:

Editor: Céline Carret

1st Editorial Decision

13 November 2015

Thank you for the submission of your manuscript to EMBO Molecular Medicine. We have now heard back from the two referees whom we asked to evaluate your manuscript. Although the referees find the study to be novel and very important, they also raise a few of concerns that must be addressed in the next version of your article.

As you will see from the comments below, both referees are supportive of your work but recommend providing explicit evidence that RA data are specific and reproducible. Both referees also would like to see more details and clarifications. We would encourage to equally provide source data as suggested by referee 2, and the gene list and data in figure 2 as supplementary information. However, we would not insist in experimentally exploring other mechanisms, discussing possibilities should be enough.

Given these evaluations, I would like to give you the opportunity to revise your manuscript, with the understanding that the referees' concerns must be fully addressed and that acceptance of the manuscript would entail a second round of review. Please note that it is EMBO Molecular Medicine policy to allow only a single round of revision and that, as acceptance or rejection of the manuscript will depend on another round of review, your responses should be as complete as possible.

EMBO Molecular Medicine has a "scooping protection" policy, whereby similar findings that are published by others during review or revision are not a criterion for rejection. Should you decide to submit a revised version, I do ask that you get in touch after three months if you have not completed

it, to update us on the status.

I look forward to receiving your revised manuscript.

***** Reviewer's comments *****

Referee #1 (Remarks):

Devalla and colleagues describe a protocol for retinoic acid (RA)-based induction of atrial cardiomyocyte differentiation and identify COUP-TF I and II as molecular regulators of atrial fate in human embryonic stem cells. They demonstrate that atrial ion channel specific compounds (vernakalant, XEN-D0101, XEN-R0703) affect only the atrial and not the ventricular cells. Finally, evidence for the utility of the atrial cell system in late preclinical drug screens is provided by demonstrating that the atrial cells provide similar information as the canine model, i.e., the standard model used by pharma for cardiovascular drug development.

Concerns:

The schematic in Fig 1A demonstrates a 10 day culture period. In the study, cells appear to be cultured for 15 and sometimes 30 days. Please, amend the schematic accordingly and also in the manuscript and Fig legends always clearly state the culture duration.

Please, provide a statement on the reproducibility of the RA protocol. I assume that Fig 1C presents data from 1 experiment.

In Fig 5 C no influence of COUP-TF I knock-down on KCNJ3 expression is shown. However, strong interaction of COUP-TFI with the KCNJ3 promoter is demonstrated (Fig 5H). Please, discuss this finding.

On page 5 a Fig S1G is introduced; this should be Fig S1F.

Fig S2 is not introduced in the main text.

Fig 5 panel G label must be corrected.

Spontaneous beating rate of the cardiomyocytes (Fig S1E) appears to be quite low. Please provide more information on the experimental conditions, in particular ambient temperature.

Referee #2 (Comments on Novelty/Model System):

This reviewer choses "inadequate" because the atrial cardiomyocytes are compared with fetal heart.

Referee #2 (Remarks):

The authors have proposed an atrial-like cardiomyocyte model through treating the hESCs by 1 $\mu\text{mol/L}$ retinoic acid (RA). Through assaying atrial- and ventricular- specific markers in the hESC-derived cardiac cells and comparing the results with that obtained from one fetal human heart, they concluded that the RA-treated hESC-derived cardiac cells were atrial-specific. The authors found that the COUP-TF transcription factors (COUP-TFI, COUP-TFII), downstream of RA, can regulate the expression of atrial-specific potassium channel genes, KCNA5 and KCNJ3. To confirm their

hypothesis that the hESC-atrial cardiomyocytes can be a suitable model for testing drugs selective for atrial cells, the authors tested the effects of antiarrhythmic agent Vernaklant, and studied the atrial-specific currents I_{Kur} and $I_{K,Ach}$ through selectively blocking and activating the atrial ion channels by 4-AP and CCh, respectively. The authors further tested their hypothesis through treating the hESC-atrial cardiomyocytes with action potential agents, XEN-D0101 and XEN-R0703. As a conclusion, they stated that the hESC-atrial cardiomyocytes are a robust model for preclinical testing to assess atrial-selectivity of novel antiarrhythmic drugs.

In general, the hypothesis and conclusion are clear. The experiments are logically designed to test their hypothesis. The hESC-atrial cardiomyocyte model should benefit the screening of ion channel-selective drugs in atrial-selective pharmacology. There are questions need should be answered before further consideration.

Major comments:

- 1) Page numbers are missing, which makes it difficult to assign specific comments.
- 2) The conclusion of section 2 of the Results part should say: "... a fetal atrial-like gene expression pattern." This comment also applies to other parts. It should be clear to the reader that the proposed cells are a model of human fetal atrial cardiomyocytes. For example, the title of the first Results paragraph should be "... promotes fetal atrial specification". Similarly, whenever it is stated in the manuscript "atrial-like", "fetal atrial-like" should be used.
- 3) As far as this reviewer can tell from the submitted data, all results about the RA-induced fate switch are relative data. Multiple different mechanisms can explain this finding. The manuscript proposes one single mechanism, i.e. selective fate decision. Another potential mechanism is selective cell death. Experiments exploring this and other alternative mechanisms should be performed wherever appropriate. For example, in the last paragraph of the first section of the Results, the second sentence should be: "... GFP expression revealed a relative decrease in the proportion of Nkx2.5 expressing cells ...". The presented version of this sentence suggests that the results in Fig. 1C are absolute numbers when they are in fact percent.
- 4) Although Fig. 1D demonstrates that a good purification of cardiomyocytes has been achieved, the number of the cells studied in Fig. S1E should be provided to explain the representation of the data.
- 5) In the second paragraph of page 9, "Knockdown of COUP-TFI or COUP-TFII in hESC-atrial CMs did not affect GFP or cTNT expression (Fig. S4F)." This reviewer could not find results related to GFP in Fig. S4F.
- 6) In the first paragraph of page 14, "These results are consistent with the finding that XEN-R0703 does not affect other membrane currents at the test concentration." The experimental data or source of the data should be provided for the comparison. A dog model was used to confirm the selectivity of XEN-R0703 predicted by hESC-atrial cardiomyocytes in vitro. Is human data to XEN-R0703 available? How the selectivity of XEN-R0703 to human predicted by hESC-atrial cardiomyocytes could be confirmed by dog model?
- 7) Figure 2D-F: This reviewer is not sure how contributory these Venn diagrams are. See next comment.
- 8) The gene list and data in Fig. 2D and 2E should be provided for the convenience of further study by the readers.

Minor comments:

- 1) In the 3rd paragraph of page 5, Fig. S1G Fig. S1F??
- 2) Fig. 4C, panels for VM should be better labeled.
- 3) Fig. 5E, 5F, label fonts should be larger; Fig. 5G Fig. 5G??
- 4) Fig. S4 C, label fonts should be larger.
- 5) Fig. S6, figures should be labeled in consistence with the caption.

1st Revision - authors' response

07 December 2014

Referee #1 (Remarks):

Devalla and colleagues describe a protocol for retinoic acid (RA)-based induction of atrial cardiomyocyte differentiation and identify COUP-TF I and II as molecular regulators of atrial fate

in human embryonic stem cells. They demonstrate that atrial ion channel specific compounds (vernakalant, XEN-D0101, XEN-R0703) affect only the atrial and not the ventricular cells. Finally, evidence for the utility of the atrial cell system in late preclinical drug screens is provided by demonstrating that the atrial cells provide similar information as the canine model, i.e., the standard model used by pharma for cardiovascular drug development.

We thank the reviewer for the comments and suggestions detailed below. We have incorporated necessary changes in the revised manuscript.

Concerns:

The schematic in Fig 1A demonstrates a 10 day culture period. In the study, cells appear to be cultured for 15 and sometimes 30 days. Please, amend the schematic accordingly and also in the manuscript and Fig legends always clearly state the culture duration.

We apologize for any confusion caused and thank the reviewer for this suggestion. We have modified the schematic (Fig. 1A) in the revised manuscript to indicate time points at which analyses were performed. Differentiation efficiency in each experiment was assessed by flow cytometry (FC) for GFP at day 15. Further characterization of EBs derived from control (CT) and RA-treated (RA) cultures was carried out by transcriptional or functional analysis between days 27-31. In the revised manuscript, we have also clearly stated timing of various experiments in the figure legends as well as in the ‘Materials and methods’ section.

Please, provide a statement on the reproducibility of the RA protocol. I assume that Fig 1C presents data from 1 experiment.

Fig. 1C was indeed from one experiment and for clarity, the legend in the revised manuscript now states, “Fig. 1C: Representative flow cytometry plots depicting percentage of GFP⁺ cells obtained at day 15, from CT and RA cultures in a typical experiment”.

The percentage of GFP⁺ cells obtained from Control or RA-treated differentiations was consistent in each experiment. In the revised manuscript, we have also included a bar graph (Fig. S1E) in the supplemental data file illustrating percentage of GFP⁺ cells in control and RA-treated differentiations obtained from $N = 3$ independent experiments; (Error bars represent SEM).

However, strong interaction of COUP-TFI with the KCNJ3 promoter is demonstrated (Fig 5H). Please, discuss this finding.

The reviewer raises an important point and we have now discussed this finding on page-17 of the revised manuscript.

“shRNA-mediated knockdown as well as ChIP experiments demonstrated that COUP-TFs regulate the atrial-selective ion channel gene, *KCNA5*. These experiments also established that *KCNJ3* and *KCNJ5* are regulated by COUP-TFII. Interestingly, although COUP-TFI showed strong interaction with the *KCNJ3* promoter, knockdown of COUP-TFI itself did not result in a decrease in the expression of *KCNJ3*. This suggests that COUP-TFI might be dispensable for the expression of *KCNJ3* and *KCNJ5* in atrial CMs. Future studies aimed at dissecting the roles of COUP-TFI and COUP-TFII during atrial differentiation are required to understand whether these genes act in synergy or possess functions independent and specific to one another. It also remains to be tested if loss of COUP-TFI can be rescued by COUP-TFII and *vice versa*. However, since myocardial ablation of COUP-TFII in the mouse results in a severe phenotype (25), it seems unlikely that COUP-TFI can account for loss of COUP-TFII. Nonetheless, it is likely that deletion of both COUP-TFI and COUP-TFII might result in a more severe phenotype”.

On page 5 a Fig S1G is introduced; this should be Fig S1F

This was an error, which has now been corrected.

Fig S2 is not introduced in the main text.

We apologize for the missing text describing Fig. S2. This is now included on page 6 of the revised manuscript.

Fig 5 panel G label must be corrected.

We thank the reviewer for noticing this error and we have corrected this in our revised manuscript.

Spontaneous beating rate of the cardiomyocytes (Fig S1E) appears to be quite low. Please provide more information on the experimental conditions, in particular ambient temperature.

The reviewer correctly noted the low spontaneous beating rate presented as Fig. S1E. This experiment was performed by counting the beating rate of $N > 10$ embryoid bodies (EBs) derived from CT or RA differentiations, but this was performed under the microscope, at room temperature. We assume that this explains the lower beating frequency. Therefore, we repeated the experiment on a platform maintained at 37 degrees. We determined the beating frequency of $N = 16$ EBs each. The rates recorded were 45/min in control EBs and 60/min in RA-treated EBs (Error bars represent S.E.M.). Please note that this is now Fig. S1F in the revised manuscript.

Referee #2 (Remarks):

The authors have proposed an atrial-like cardiomyocyte model through treating the hESCs by 1 µmol/L retinoic acid (RA). Through assaying atrial- and ventricular- specific markers in the hESC-derived cardiac cells and comparing the results with that obtained from one fetal human heart, they concluded that the RA-treated hESC-derived cardiac cells were atrial-specific. The authors found that the COUP-TF transcription factors (COUP-TFI, COUP-TFII), downstream of RA, can regulate the expression of atrial-specific potassium channel genes, KCNA5 and KCNJ3. To confirm their hypothesis that the hESC-atrial cardiomyocytes can be a suitable model for testing drugs selective for atrial cells, the authors tested the effects of antiarrhythmic agent Vernaklant, and studied the atrial-specific currents I_{Kur} and $I_{K,Ach}$ through selectively blocking and activating the atrial ion channels by 4-AP and CCh, respectively. The authors further tested their hypothesis through treating the

hESC-atrial cardiomyocytes with action potential agents, XEN-D0101 and XEN-R0703. As a conclusion, they stated that the hESC-atrial cardiomyocytes are a robust model for preclinical testing to assess atrial-selectivity of novel antiarrhythmic drugs.

In general, the hypothesis and conclusion are clear. The experiments are logically designed to test their hypothesis. The hESC-atrial cardiomyocyte model should benefit the screening of ion channel-selective drugs in atrial-selective pharmacology. There are questions need should be answered before further consideration.

We thank the reviewer for appreciating our work and we have addressed all the questions raised.

Major comments:

1) Page numbers are missing, which makes it difficult to assign specific comments.

We apologize for this. We are not quite sure how this occurred, since in the version uploaded for review, page numbers were indicated. We also checked the final merge document and page numbers were present. We have now uploaded a '.doc' file (instead of a '.docx' file) and page numbers are indicated on the top left of the revised manuscript. We hope that this solves the problem.

2) The conclusion of section 2 of the Results part should say: "... a fetal atrial-like gene expression pattern." This comment also applies to other parts. It should be clear to the reader that the proposed cells are a model of human fetal atrial cardiomyocytes. For example, the title of the first Results paragraph should be "... promotes fetal atrial specification". Similarly, whenever it is stated in the manuscript "atrial-like", "fetal atrial-like" should be used.

The reviewer raises an important point. Current cardiac differentiation protocols of hPSCs do indeed result in cardiomyocytes that resemble human fetal cardiomyocytes rather than their adult counterparts. Fetal human tissue was therefore used as a comparison for global transcriptional analysis of hESC-atrial and hESC-ventricular CMs. As suggested by the reviewer, we have changed the sentence on page-7 to "Therefore, the transcriptional profile of RA+ CMs suggested a fetal atrial-like gene expression pattern compared with control CMs, which expressed higher levels of ventricular transcripts".

However, for the electrophysiology experiments, due to limitations in availability of human fetal atrial cells, we did not directly compare hESC-atrial CMs to fetal cells to confirm that these cells are fetal-like. Hence, to avoid confusion, we referred to these cells as atrial-like CMs throughout the manuscript. Additionally, we have mentioned in the discussion on page-18 that immaturity of hESC-atrial CMs does not preclude their use from pharmacological screening assays since they respond to atrial-selective drugs.

“hPSC-derived CMs are spontaneously active and exhibit depolarized RMP. Evidence suggests that they are developmentally immature, resembling human fetal CMs rather than their adult counterparts (47). In our study, RMP did not differ between hESC-atrial CMs and hESC-ventricular CMs indicating similar level of maturity in both the groups. Nonetheless, hESC-atrial CMs respond to atrial-selective ion channel blockers demonstrating that an immature electrical phenotype does not preclude their use in preclinical drug screening and pharmacology”.

3) As far as this reviewer can tell from the submitted data, all results about the RA-induced fate switch are relative data. Multiple different mechanisms can explain this finding. The manuscript proposes one single mechanism, i.e. selective fate decision. Another potential mechanism is selective cell death. Experiments exploring this and other alternative mechanisms should be performed wherever appropriate. For example, in the last paragraph of the first section of the Results, the second sentence should be: "... GFP expression revealed a relative decrease in the proportion of Nkx2.5 expressing cells ...". The presented version of this sentence suggests that the results in Fig. 1C are absolute numbers when they are in fact percent.

The reviewer is correct that multiple mechanisms might contribute to subtype specification towards atrial or ventricular cardiomyocytes. This is discussed in the revised manuscript on page-16.

We found that COUP-TFs are markedly upregulated in response to retinoic acid and thus hypothesized a role for these genes in atrial differentiation. Although it is possible that programmed cell death leading to selective survival could be a part of the mechanism, we did not actually observe any unusual apoptosis in RA-treated EBs compared to controls. Rapid and robust induction of COUP-TFs also suggested transcriptional regulation rather than mere loss of CMs. Furthermore, differentiating EBs during early stages of differentiation are comprised of multiple cell types. In order to study the possibility of selective cell death in response to RA during these early stages of differentiation, a dual reporter line enabling the tracking of a cardiac progenitor marker (expressed by presumptive ventricular cells) as well as an apoptosis marker would facilitate the identification of which cell types (if any), are directed towards cell death. Earlier studies in amniotes have suggested a possibility that RA is required for the formation of atrial cardiomyocytes and that in its absence, cardiac precursors differentiate to ventricular cells (Hochgreb et al., *Development* 2003, 130:5363–5374; Simoes-Costa et al., *Devl Biol* 2000, 277:1-15). However, a recent study in zebrafish has shown that RA signaling acts via different mechanisms to limit both atrial and ventricular cell numbers but not at the expense of each other (Waxman et al., *Dev Cell* 2008, 15(6):923-34). Extensive studies in mammalian model systems are thus required to resolve these differences and identify the precise role of RA in specification and sizing of the cardiac chambers.

The reviewer rightly points out that it would be better to use the word ‘proportion’ instead of ‘number’. This sentence has been corrected in the revised manuscript and now reads, “Flow cytometry analysis of GFP expression revealed a decrease in the proportion of NKX2.5 expressing cells upon treatment with RA”.

In the revised manuscript, we have also included a bar graph (Fig. S1E) in the supplemental data file illustrating percentage of GFP+ cells in control and RA-treated differentiations ($N = 3$ independent experiments; Error bars represent SEM).

4) Although Fig. 1D demonstrates that a good purification of cardiomyocytes has been achieved, the number of the cells studied in Fig. S1E should be provided to explain the representation of the data.

In the revised manuscript, Fig. S1F (Fig. S1E in the earlier version of the manuscript) is a representation of $N=16$ EBs each, counted on a platform maintained at 37 degrees.

5) In the second paragraph of page 9, "Knockdown of COUP-TFI or COUP-TFII in hESC-atrial CMs did not affect GFP or cTNT expression (Fig. S4F)." This reviewer could not find results related to GFP in Fig. S4F.

We apologize for this omission and have now included an additional figure (Fig.S5), in the revised manuscript displaying GFP images following knockdown of COUP-TFI or COUP-TFII.

6) In the first paragraph of page 14, "These results are consistent with the finding that XEN-R0703 does not affect other membrane currents at the test concentration." The experimental data or source of the data should be provided for the comparison. A dog model was used to confirm the selectivity of XEN-R0703 predicted by hESC-atrial cardiomyocytes in vitro. Is human data to XEN-R0703 available? How the selectivity of XEN-R0703 to human predicted by hESC-atrial cardiomyocytes could be confirmed by dog model?

We apologize for not clearly describing the experimental findings with XEN-R0703. We have included experimental data for XEN-R0703 in the supplemental data file (Table S5; Fig. S7-8). Furthermore, we have tested the effect of XEN-R0703 on $I_{K_{ACh}}$ in isolated human atrial myocytes (Fig. S8).

“The results observed with XEN-R0703 in hESC-atrial CMs are consistent with the finding that this drug does not affect other membrane currents at the tested concentration”. This conclusion was reached on the basis of experimental effects of XEN-R0703 on various membrane currents expressed in HEK293 or CHO cells. As summarized in Table S5 of the supplement, the IC₅₀ for Kir3.1/3.4 (which encode the acetylcholine-activated current I_{K,ACh}), is in the nanomolar range, while it is in the micromolar range for other important depolarizing and repolarizing cardiac membrane currents such as hERG or K_{ir}2.1.

Table S5. Summary of ion channel pharmacology of XEN-R0703

Ion Channel	Platform	IC ₅₀	K _{ir} 3.1/3.4 Selectivity Ratio
Kir3.1/3.4	CP	59 nM	-
hERG	CP	5.7μM	~100
Na _v 1.5 (1Hz)	QPatch	19μM	>300
Nav1.5 (Tonic)	QPatch	22μM	~300
Ca _v 1.2	Flex	30μM	~500
K _{ir} 2.1	CP	>3μM	>>50
I _{K,ACh}	Myocyte /CP	81 % @ 300nM	

Due to limitations in obtaining good preparations of human atrial myocytes, extensive experiments such as effects of XEN-R0703 on action potentials could not be recorded. However, we were able to study the effect of XEN-R0703 on I_{K,ACh} current in freshly isolated human atrial myocytes and found that 300 nM XEN-R0703 blocked I_{K,ACh} by approximately 80% (Fig. S8 of the supplement). Please note that we now have included the number of cells used.

Based on these experiments, we tested the effect of XEN-R0703 on action potentials of hESC-atrial and hESC-ventricular myocytes and expected to see an effect only in hESC-atrial CMs. As summarized in Fig. 8B and Table 6 of the supplement, XEN-R0703 reversibly depolarized the resting membrane potential and restored the action potential shortening caused by Carbachol in hESC-atrial CMs. By contrast, XEN-R0703 did not have any effect on the action potentials of hESC-ventricular CMs indicating the absence of I_{K,ACh} in these cells. Furthermore the absence of XEN-R0703 effects in hESC-ventricular CMs demonstrated that the drug had no effect on other membrane currents. This is in agreement with the much higher IC₅₀ values for other cardiac membrane currents (Table 5 of the supplement).

Next, we tested if the atrial-selectivity of XEN-R0703 predicted in hESC-atrial CMs could be confirmed in an existing pre-clinical model used in the pharmaceutical industry. The dog was chosen for this experiment since they are routinely used in the pharmaceutical industry to assess cardiac safety and the risk of drug-induced ventricular arrhythmias (Olson et al., Regul Toxicol Pharmacol. 2000; 32: 56–67). The dog is also regarded as one of the most predictive animal species of human cardiovascular toxicity. To this effect, we tested the effect of XEN-R0703 in a RAP dog model of persistent AF.

We found, as depicted in Fig. 8D, that while XEN-R0703 increased right atrial refractory period (AERP) in a dose dependent manner, it did not affect the QTc interval. This indicates that the drug does not have an adverse effect on the ventricle indicating selective effect on the atria. Thus, using an *in vivo* dog model, we were able to validate that hESC-atrial CMs predict atrial selectivity

We have rephrased the text on page 14 of the revised manuscript to describe this more clearly. “These results also demonstrate that XEN-R0703 does not affect other membrane currents present in hESC-ventricular CMs, consistent with the effects of the drug observed in HEK-293 cells

expressing various ion channels (Table S5, Fig. S7)". We have also added a small paragraph discussing these results on page-18 of the revised manuscript.

7) Figure 2D-F: This reviewer is not sure how contributory these Venn diagrams are. See next comment.

We have used Venn diagrams in Fig. 2 to indicate (at a quick glance) the shift in gene expression between hESC-atrial and hESC-ventricular CMs. We have now modified these panels (2D-E) to improve visual representation of the data.

8) The gene list and data in Fig. 2D and 2E should be provided for the convenience of further study by the readers.

We have now included the gene lists from Fig. 2D and 2E as an additional supplemental data file (File F3).

Minor comments:

1) In the 3rd paragraph of page 5, Fig. S1G→Fig. S1F??

We apologize for this error and have now corrected this in our revised manuscript.

2) Fig. 4C, panels for VM should be better labeled.

We thank the reviewer for noticing the missing labels for the VM panel in Fig. 4C. We have now included the labels in the revised manuscript.

3) Fig. 5E, 5F, label fonts should be larger; Fig. 5GC→Fig. 5G??-

We apologize for the smaller font of the labels in schematics Fig. 5E and 5F in the earlier version of the manuscript. We have now increased the font size and hope that it is legible. We have also corrected the error in labeling of Fig. 5G.

4) Fig. S4 C, label fonts should be larger.

We have increased the size of this image in the revised manuscript.

5) Fig. S6, figures should be labeled in consistence with the caption.

We apologize for the missing labels in the earlier version of the manuscript. This has now been corrected and it is now Fig. S7 in the revised version.

Thank you for the submission of your revised manuscript to EMBO Molecular Medicine. We have now received the enclosed report from the referee asked to re-assess it. As you will see the reviewer is now supportive and I am pleased to inform you that we will be able to accept your manuscript pending the following final amendments:

1) please address carefully all remaining issues from this referee and reply in a rebuttal letter to be provided as a doc file.

Please submit your revised manuscript In January 2015. I look forward to seeing a revised form of your manuscript as soon as possible.

***** Reviewer's comments *****

Referee #2 (Remarks):

In the revised manuscript and the responding letter, the comments of the reviewers have been addressed. This reviewer feels that the manuscript should be publishable if the following questions are properly addressed:

1. The apoptosis/selective cell death question was not addressed with experiments. Although the authors stated that they did not see apoptosis, no results were provided.
2. Page 4, paragraph 4, EBs embryoid bodies (EBs)?
3. Page 5, paragraph 2, MYL2 (Fig. S1C), appears in Fig. S1C as MLC2V, please make sure they are consistent.
4. Fig. 2D, UP upregulated? Abbreviations should be noted in the caption.
5. Fig. S3D looks similar with Fig. S3H. Were those images acquired from separately stained slices sectioned from the same heart sample, or the same slice multi-stained by COUP-TFI, COUP-TFII and TNNT3 antibodies? Please describe the acquiring procedure of those images.

Rebuttal to referee-2

1. The apoptosis/selective cell death question was not addressed with experiments. Although the authors stated that they did not see apoptosis, no results were provided.

In our initial experiments directing atrial differentiations with retinoic acid, we did not see any indication that there is increased or decreased cell death in RA-treated embryoid bodies. Moreover, differentiating EBs during early stages of differentiation are comprised of multiple cell types. In order to accurately study the possibility of selective cell death in response to RA during these early stages of differentiation, a dual reporter line enabling the tracking of a cardiac progenitor marker (expressed by presumptive ventricular cells) as well as an apoptosis marker would facilitate the identification of which cell types (if any), are directed towards cell death. Unfortunately, these tools are not available and experiments suggested by the referee would be interesting as a follow-up study. An additional point to consider is the time when apoptosis may occur. If we would demonstrate no difference in the level of apoptosis during differentiation at a specific time point, we would still not be able to exclude that this is not the case at an earlier or later stage than initially investigated. Preparation and differentiation of cells under control and RA-treated conditions, followed by quantitative analysis will most likely take several months and is out of scope for the current study.

Hence, due to 1) lack of any indication that there is increased or decreased cell death in RA-treated EBs compared to controls and 2) lack of appropriate tools to draw definitive conclusions, we did not perform additional experiments to further investigate selective cell death.

2. Page 4, paragraph 4, EBs; embryoid bodies (EBs)?

We apologize for not expanding the abbreviation of EBs in version-2 of our manuscript. This has now been corrected on page-4 of the revised manuscript.

3. Page 5, paragraph 2, MYL2 (Fig. SIC), appears in Fig. SIC as MLC2V, please make sure they are consistent.

Thank you for bringing this to our notice and it has now been corrected on page-5 of the revised manuscript.

4. Fig. 2D, UP; upregulated? Abbreviations should be noted in the caption.

The legend for Fig.2 in the revised manuscript states ‘UP’ in parentheses and now reads, “Venn diagram to illustrate overlap of gene lists upregulated (UP) in **(D)** CT+ CMs and **(E)** RA+ CMs with genes expressed in atria and ventricles of 15 week-old fetal heart”.

5. Fig. S3D looks similar with Fig. S3H. Were those images acquired from separately stained slices sectioned from the same heart sample, or the same slice multi-stained by COUP-TFI, COUP-TFII and TNNI3 antibodies? Please describe the acquiring procedure of those images.

We apologize that this was not clearly stated in the expanded methods. Serial sections of the same heart were used for immunostaining of COUP-TFI or COUP-TFII. Sister sections were thus separately stained for COUP-TFI+TNNI3 or COUP-TFII+TNNI3. In the revised manuscript, methods on page-4 of ‘Expanded view material’ now states this information.